# TAGAP instructs Th17 differentiation by bridging Dectin activation to EPHB2 signaling in innate antifungal response

Jianwen Chen[1,13], Ruirui He[1,13], Wanwei Sun[1✉], Ru Gao[1], Qianwen Peng[1], Liwen Zhu[2], Yanyun Du[1], Xiaojian Ma[1], Xiaoli Guo[1], Huazhi Zhang[1], Chengcheng Tan[1], Junhan Wang[3], Wei Zhang[4], Xiufang Weng[5], Jianghong Man[6], Hermann Bauer[7], Qing K. Wang[1,8,9], Bradley N. Martin [10], Cun-Jin Zhang [2✉], Xiaoxia Li [11] & Chenhui Wang [1,12✉]

The *TAGAP* gene locus has been linked to several infectious diseases or autoimmune diseases, including candidemia and multiple sclerosis. While previous studies have described a role of TAGAP in T cells, much less is known about its function in other cell types. Here we report that TAGAP is required for Dectin-induced anti-fungal signaling and proinflammatory cytokine production in myeloid cells. Following stimulation with Dectin ligands, TAGAP is phosphorylated by EPHB2 at tyrosine 310, which bridges proximal Dectin-induced EPHB2 activity to downstream CARD9-mediated signaling pathways. During *Candida albicans* infection, mice lacking TAGAP mount defective immune responses, impaired Th17 cell differentiation, and higher fungal burden. Similarly, in experimental autoimmune encephalomyelitis model of multiple sclerosis, TAGAP deficient mice develop significantly attenuated disease. In summary, we report that TAGAP plays an important role in linking Dectin-induced signaling to the promotion of effective T helper cell immune responses, during both antifungal host defense and autoimmunity.

[1] Key Laboratory of Molecular Biophysics of the Ministry of Education, National Engineering Research Center for Nanomedicine, College of Life Science and Technology, Huazhong University of Science and Technology, Wuhan 430074, China. [2] Department of Neurology of Drum Tower Hospital, Medical School and the State Key Laboratory of Pharmaceutical Biotechnology, Nanjing University, Nanjing, Jiangsu 210008, China. [3] University-Affiliated Hospital, Huazhong University of Science and Technology, Wuhan 430074, China. [4] Hepatic Surgery Center, Tongji Hospital, Tongji Medical College, Huazhong University of Science and Technology, Wuhan, China. [5] Department of Immunology, School of Basic Medicine, Tongji Medical College, Huazhong University of Science and Technology, Wuhan 430030, China. [6] State Key Laboratory of Proteomics, Institute of Basic Medical Sciences, National Center of Biomedical Analysis, Beijing 100850, China. [7] Department of Developmental Genetics, Max Planck Institute for Molecular Genetics, Ihnestr. 63-73, 14195 Berlin, Germany. [8] Department of Molecular Cardiology, Lerner Research Institute, Cleveland Clinic Lerner College of Medicine of Case Western Reserve University, Cleveland, OH, USA. [9] Department of Molecular Medicine, Department of Genetics and Genome Science, Cleveland Clinic Lerner College of Medicine of Case Western Reserve University, Cleveland, OH, USA. [10] Department of Medicine, Brigham and Women's Hospital, Harvard Medical School, Boston, MA, USA. [11] Department of Inflammation and Immunity, Lerner Research Institute, Cleveland Clinic, Cleveland 44106, USA. [12] Wuhan Institute of Biotechnology, Wuhan, Hubei 430070, China. [13]These authors contributed equally: Jianwen Chen, Ruirui He. ✉email: 2018501003@hust.edu.cn; zhangcj@nju.edu.cn; wangchenhui@hust.edu.cn

T helper cells, such as Th1 and Th17 cells, play a pivotal role in the host defense against various pathogens, including bacteria and fungi. In recent years, however, increasing evidence has shown that Th17 and Th1 cells also play a central role in the initiation and pathogenesis of many autoimmune diseases, including multiple sclerosis (MS), inflammatory bowel disease (IBD), rheumatoid arthritis (RA), and psoriasis, among others[1,2]. The discovery of IL-23a, which shares a p40 chain with IL-12a, led to the discovery of Th17 cells[3]. Early studies found that IL-23a deficiency, but not IL-12a deficiency, protected mice from developing experimental encephalomyelitis (EAE) and collagen-induced arthritis (CIA)[4]. Indeed, later studies implicated Th17 cells in the pathogenesis of many autoimmune diseases, both in mouse models and in human diseases[1–5].

In response to fungal and bacterial infections, various pattern recognition receptors, such as C-type lectin receptors (CLRs), Toll-like receptors (TLRs), and Nod-like receptors (NLRs), initiate the host immune defense against invading pathogens[6,7]. CLRs, such as Dectins and Mincle, respond to fungal products and induce Th17 cell polarization to promote fungi clearance. The fungal cell wall components β-glucan, α-mannan, and glycolipid are detected by Dectin-1, Dectin-2, Dectin-3, and Mincle, which recruit and activate spleen tyrosine kinase (Syk) to induce CARD9–BCL10–MALT1 complex-dependent NF-κB signaling activation in macrophages or dendritic cells (DCs), which then leads to the release of proinflammatory cytokines and chemokines, such as IL-23a, IL-12a, IL-1β, TNFα, and CXCL1[8–10]. In addition to the NF-κB signaling pathway, stimulation by fungal and bacterial components can also activate the MAPK and NFAT pathways, which are also believed to play an essential role in the pathogen-killing process[11,12]. The released cytokines, such as IL-1β, IL-6, and IL-23a, have a pivotal role in T-cell differentiation into Th17 cells, whereas IL-12a can drive Th1 cell differentiation[1,2]. The cytokines and chemokines that are secreted by activated Th17 cells, such as IL-17A and IL-17F, further trigger IL-17 signaling pathways to recruit neutrophils, and distinguish invading pathogens. However, dysregulation of T helper cell abundance, especially Th17 cells can result in susceptibility to developing many autoimmune diseases[13–16]. Genome-wide association studies (GWAS) have identified a number of genes conferring susceptibility to autoimmune and infectious diseases; however, the specific biological function of many of these genes remains unclear. Elucidation of the mechanism(s) governing genetic susceptibility to complex disease states, such as autoimmune disease, may provide novel targets for therapeutic development.

T-cell activation Rho GTPase-activating protein (TAGAP) is a GTPase-activating protein, and many single nucleotide polymorphisms (SNPs) near or within the TAGAP gene have been found to be associated with susceptibility to many autoimmune diseases and infectious diseases, including MS, Crohn's disease, psoriasis, RA, celiac disease, and candidemia[17–21]. TAGAP protein is a GAP domain containing protein, and previous study found that TAGAP has a role in T-cell differentiation[22,23]. Here, we report that TAGAP is required for Dectin-1, Dectin-2/3 and mincle ligands-induced signaling pathway activation and proinflammatory induction in macrophages. We provide evidence that TAGAP functions as an adaptor to mediate upstream EPHB2 and downstream CARD9 signaling, leading to the activation of various CLR pathways. Mechanically, EPHB2 is phosphorylated by Syk after Dectin ligands stimulation, and further phosphorylates TAGAP at the site of Y310. Phosphorylated TAGAP at site of Y310 recruits CARD9 for the downstream signal transduction. Owing to the defective production of proinflammatory cytokines, such as IL-23a and IL-12a, in response to stimulation by Dectin ligands, TAGAP-deficient mice have decreased Th17 and Th1 cell populations, and are susceptible to Candida albicans infection. TAGAP-deficient mice also have a much less severe myelin oligodendrocyte glycoprotein (MOG$_{35–55}$)–induced EAE phenotype compared with control mice. In addition, we find dysregulated Th17 and Th1 cell populations in PBMC samples from individuals who carry TAGAP human disease associated variants, as well as a positive correlation between TAGAP mRNA expression level and Th17 cell abundance in the PBMCs. Finally, we show that the broad-spectrum tyrosine kinase inhibitors dasatinib and vandetanib can block Th17 and Th1 cell polarization, and greatly reduce mice EAE severity by inhibiting Th17 and Th1 differentiation in vivo, which suggests that these two existing drugs could be used to treat autoimmune diseases such as MS. In summary, we report that TAGAP has an important role in macrophages, linking membrane-proximal Dectin-induced antifungal signaling to the promotion of effective T helper cell immune responses, during both antifungal host defense and autoimmunity.

## Results

**TAGAP is required for antifungal signaling pathway activation in macrophages.** To understand the functional role of TAGAP in vivo, we first examined TAGAP mRNA expression in different mouse tissues. Consistent with data from the gene expression database BioGPS (http://biogps.org/#goto=genereport&id=117289), TAGAP was mainly expressed in peripheral blood mononuclear cells (PBMCs) and in the spleen. Macrophages expressed the highest levels of TAGAP out of all of the hematological cells tested (Fig. 1a).

A previous study found that TAGAP is involved in HSV-mediated induction of the TLR3 pathway[24], and we first explored the function of TAGAP in macrophages by examining TLR3 or TLR4 ligands-induced signaling pathway activation in bone marrow-derived macrophages (BMDMs) from heterozygous control mice or TAGAP-deficient mice. We did not find any significant difference in terms of signaling activation and gene expression between BMDMs from heterozygous control mice or TAGAP-deficient mice after TLR3 and TLR4 ligands stimulation (Supplementary Fig. 1A–D). A SNP in TAGAP is associated with the susceptibility to candidemia[21], which suggests that TAGAP might be involved in antifungal pathways, so we explored whether TAGAP deficiency would affect antifungal signaling pathway activation in macrophages. Interestingly, Dectin-1 ligand Curdlan-induced NF-κB and MAPK pathway signaling was greatly reduced, and proinflammatory gene expression was also markedly attenuated in TAGAP-deficient BMDMs compared with control cells (Fig. 1b, c). Similar to the results obtained from Curdlan stimulation, heat-killed C. albicans and D-zymosan-induced NF-κB and MAPK activation and proinflammatory gene expression were also significantly diminished (Fig. 1d, e). Consistent with the findings in mouse cells, signaling activation and proinflammatory gene induction by heat-killed C. albicans were almost abolished in TAGAP knocked down THP-1 cells (Fig. 1f). Interestingly, phosphorylation of upstream kinases Raf-1 and Syk of Dectin pathway did not have significant defect in TAGAP-deficient BMDMs compared with that in control cells after heat-killed C. albicans stimulation, which suggests that TAGAP functions downstream of Raf-1 and Syk in the Dectin-1 ligand-induced signaling pathway (Fig. 1g). Together, these data indicate that TAGAP has a critical role in Dectin-1 ligand-induced signaling activation and proinflammatory cytokines expression.

To identify transcriptome changes in TAGAP-deficient cells compared with that of control cells, we performed RNA-Sequencing to identify the differentially induced genes between BMDMs from heterozygous control mice or TAGAP-deficient

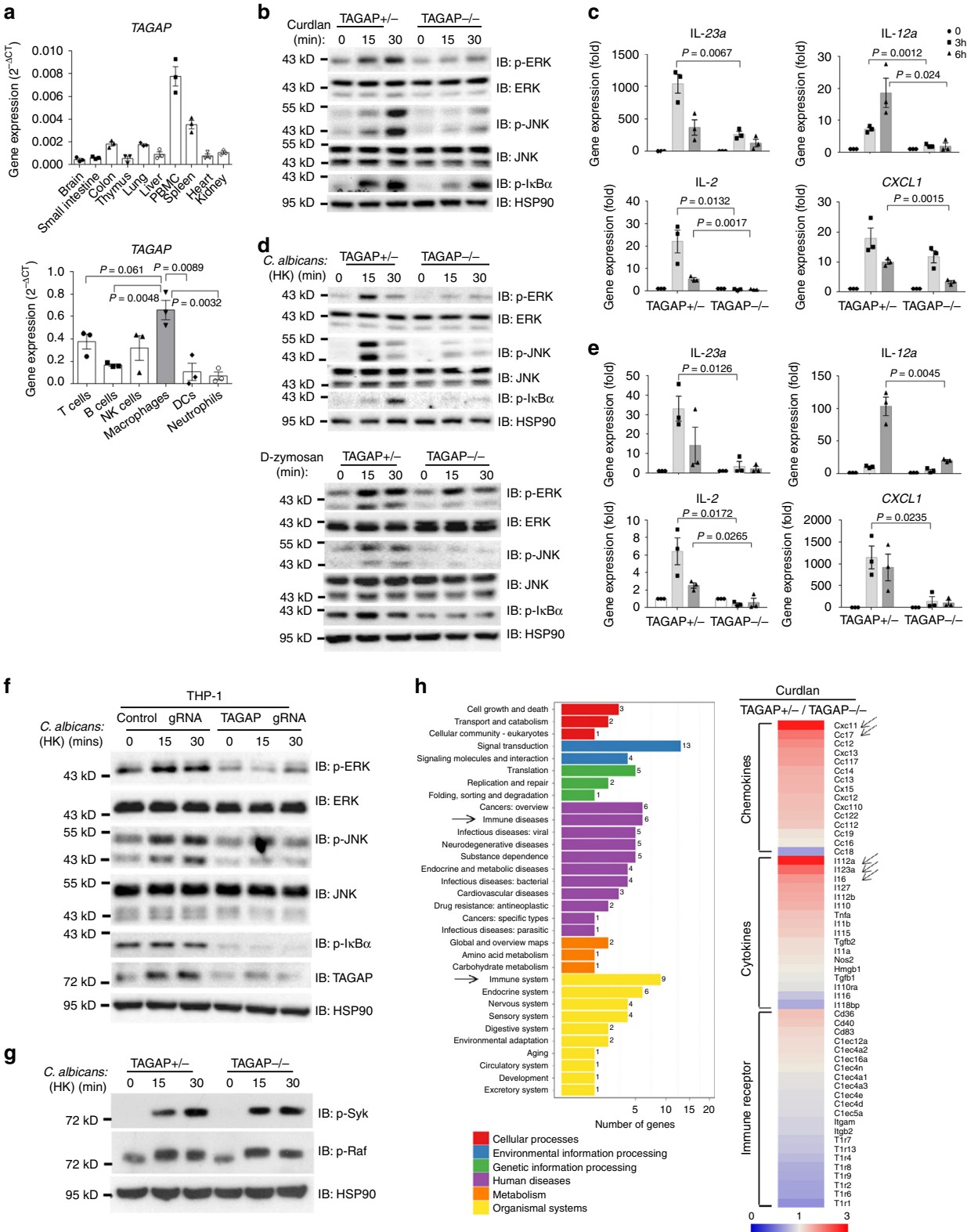

mice after Curdlan stimulation. KEGG pathway analysis revealed that the genes whose expression levels were most changed were involved in immune disease and immune system (Fig. 1h). The cytokines and chemokines which were most affected by TAGAP deficiency were *IL-12a*, *IL-23a*, *IL-6*, *CXCL1*, and *CCL7* (Fig. 1h), which correlates very well with our earlier results (Fig. 1c, e).

Dectin-2/3 receptors recognizes the fungal cell wall component α-mannan and fungal hyphae, and are pivotal for proinflammatory gene induction and T-cell priming[7,9]. Stimulation with α-mannan resulted in greatly reduced gene induction and signaling pathway activation in TAGAP-deficient BMDMs compared with that from control cells (Fig. 2a, b). Consistently, signaling

**Fig. 1 TAGAP is required for Dectin-1 ligand-induced signaling activation. a** Real-time PCR was done from different organs (upper panel) and cell types (lower panel) from three mice, and the result was shown. **b, c** BMDMs from heterozygous control mice or TAGAP-deficient mice were stimulated with Curdlan (100 μg/ml) for the indicated times, followed by western blot or real-time PCR analysis of indicated proteins and gene expression. **d** BMDMs from heterozygous control or TAGAP-deficient mice were stimulated with heat-killed *C. albicans* sc-5314 (upper panel, MOI = 2) or D-zymosan (lower panel, 100 μg/ml) for the indicated times, followed by western blot analysis of indicated proteins. **e** BMDMs from heterozygous control or TAGAP-deficient mice were stimulated with heat-killed *C. albicans* sc-5314 for 3 and 6 h, followed by real-time PCR analysis of indicated gene expression. **f** Control- gRNA or TAGAP-gRNA knocked down THP-1 cells were stimulated with heat-killed *C. albicans* sc-5314 (MOI = 2) for indicated times, followed by western blot analysis of indicated proteins. **g** BMDMs from heterozygous control mice or TAGAP-deficient mice were stimulated with heat-killed *C. albicans* sc-5314 for indicated times, followed by western blot analysis of indicated proteins. **h** BMDMs from heterozygous control mice or TAGAP-deficient mice were stimulated with Curdlan (100 μg/ml), followed by RNA-Seq analysis of gene expression. Heat-map of RNA-Seq data was shown. Arrow indicated top-changed gene expression in cytokine and chemokine groups. The scale bar representing fold induction was shown. *$P < 0.05$; **$P < 0.01$; ***$P < 0.001$ based on two-sided unpaired *t* test for panel **a**, **c**, **e**. All error bars represent SEM of technical replicates. Data are representative of three independent experiments except for **h**.

activation induction by α-mannan was also greatly reduced in TAGAP knocked down THP-1 cells (Fig. 2c). TAGAP-deficient BMDMs stimulated with the mincle ligand TDB also showed defects in gene expression and signaling pathway activation (Fig. 2d, e). Together, these data indicate that TAGAP plays a critical role in Dectin-2/3 and mincle ligands-induced antifungal signaling pathway activation.

Consistent with the finding in BMDMs, TAGAP-deficient DCs also had defect in NF-κB and MAPK activations after Dectin-1 ligand D-zymosan stimulation (Supplementary Fig. 2A). Proinflammatory cytokines expression was decreased in TAGAP-deficient BMDCs compared with that from control cells after D-zymosan stimulation (Supplementary Fig. 2B). TAGAP-deficient DCs also showed defect signaling activation and proinflammatory cytokines expression after Dectin-2/3 ligand α-Mannan stimulation (Supplementary Fig. 2C, D). Although after TLR3/4 ligands Poly(I:C) or LPS stimulation, signaling activation and proinflammatory gene expression did not show any significant defect in TAGAP-deficient DCs compared with that in control cells, which was consistent with the finding in BMDMs (Supplementary Fig. 2E, F). Overall, these data indicate that TAGAP also has a critical role in the antifungal pathway activation in DCs.

**TAGAP-binding partner EPHB2 is indispensable for antifungal signaling activation**. To understand how TAGAP is involved in the antifungal signaling pathway, we first examined the interaction between Dectin-1 and TAGAP or a GAP domain deletion mutant of TAGAP (TAGAPΔGAP). Interestingly, TAGAP can bind Dectin-1, whereas TAGAPΔGAP cannot, which indicates that the GAP domain of TAGAP is essential for the recruitment of TAGAP to Dectin-1 (Fig. 3a). TAGAP also bound to Dectin-2, whereas TAGAPΔGAP did not (Supplementary Fig. 3A). As the TAGAP protein does not contain an ITAM domain or a SH2 domain[25], the interaction between Dectin-1 and TAGAP is most likely mediated by other proteins. We therefore performed immunoprecipitation and mass spectrum analysis to identify TAGAP-interacting proteins. Two family proteins were identified: syntrophin and Eph receptor family proteins. We focused on EPHB2, as this protein was the most abundant protein identified by mass spectrometry (Fig. 3b). In human monocytes cell lines THP-1 and U937, EPHB2 protein was strongly induced by PMA, which suggested that EPHB2 may have an important role in the macrophages (Fig. 3c and Supplementary Fig. 3B). Next, we confirmed the interaction between TAGAP and EPHB2 by co-immunoprecipitation, and found that wild-type TAGAP interacted with EPHB2 whereas TAGAPΔGAP mutant showed greatly reduced binding to EPHB2, which suggests that the GAP domain of TAGAP has an important role in mediating the binding to EPHB2 (Fig. 3d). We then mapped the EPHB2 interaction domain for TAGAP binding, and found that

the SAM domain of EPHB2 is the most important region for the binding to TAGAP (Fig. 3e). Next, we explore whether EPHB2 can bind Dectins, and found that EPHB2 bound both Dectin-1 and Dectin-2 (Supplementary Fig. 3C, D). The fact that EPHB2 can bind both Dectin-1/2 and TAGAP suggests that the interaction between Dectin-1/2 and TAGAP may be mediated by EPHB2. To explore this possibility, we generated EPHB2-knocked down 293 T cell by CRISPR-Cas9, and found that the interaction between TAGAP and Dectin-1/Dectin-2 was significantly decreased in EPHB2-knocked down cells compared with that in control 293 T cells (Supplementary Fig. 3E, F). The residual interaction between TAGAP and Dectin-1 and Dectin-2 in the EPHB2-knocked down cells may be mediated by the residual EPHB2, or there may be other molecule which also mediates their interaction. We could not detect any interaction between TAGAP and Dectin-3, which indicates that the functional role of TAGAP in Dectin-2/3 pathway is mainly through Dectin-2 (Supplementary Fig. 3F). These data indicate that the interaction between TAGAP and Dectin-1/2 is mainly mediated through EPHB2.

Next, we found that proinflammatory gene induction was greatly reduced in EPHB2 knockdown BMDMs compared with that in control knocked down cells after Curdlan stimulation (Fig. 3f). Consistently, heat-killed *C. albicans*, Curdlan and α-Mannan-induced NF-κB and MAPK activation was greatly reduced in EPHB2 guide RNA (gRNA)-infected THP-1 cells compared with that in control gRNA infected cells (Fig. 3g–i). Proinflammatory gene expression were also attenuated in EPHB2-gRNA-infected THP-1 cells compared with that in control cells after heat-killed *C. albicans* stimulation (Fig. 3j). These data indicate that EPHB2 has a critical role in the antifungal signaling pathways.

**EPHB2 phosphorylates TAGAP at site of tyrosine 310.** As EPHB2 is a tyrosine kinase, similar to Syk, we explored whether EPHB2 can be phosphorylated after Dectin ligand stimulation. Interestingly, EPHB2 was phosphorylated in human THP-1 cells after stimulation with heat-killed *C. albicans* (Fig. 4a). EPHB2 was previously found to bind to Syk in the cells[26], and this interaction was confirmed by co-immunoprecipitation (Fig. 4b). Based on the results from the in vitro kinase assay, EPHB2 can be phosphorylated by wild-type Syk but not a kinase dead (KD) Syk mutant, which indicates that EPHB2 functions downstream of Syk in the Dectin-1 pathway (Fig. 4c). Consistent with this finding, the interaction between Dectin-1 and EPHB2 was greatly increased by co-expression with wild-type Syk compared with co-expression with a KD (K402R) Syk mutant (Fig. 4d). After Syk inhibitor Piceatannol treatment, phosphorylated EPHB2 was greatly attenuated after heat-killed *C. albicans* stimulation, which indicates that endogenous EPHB2 phosphorylation after Dectin-1 ligand stimulation is dependent on Syk (Supplementary Fig. 3G).

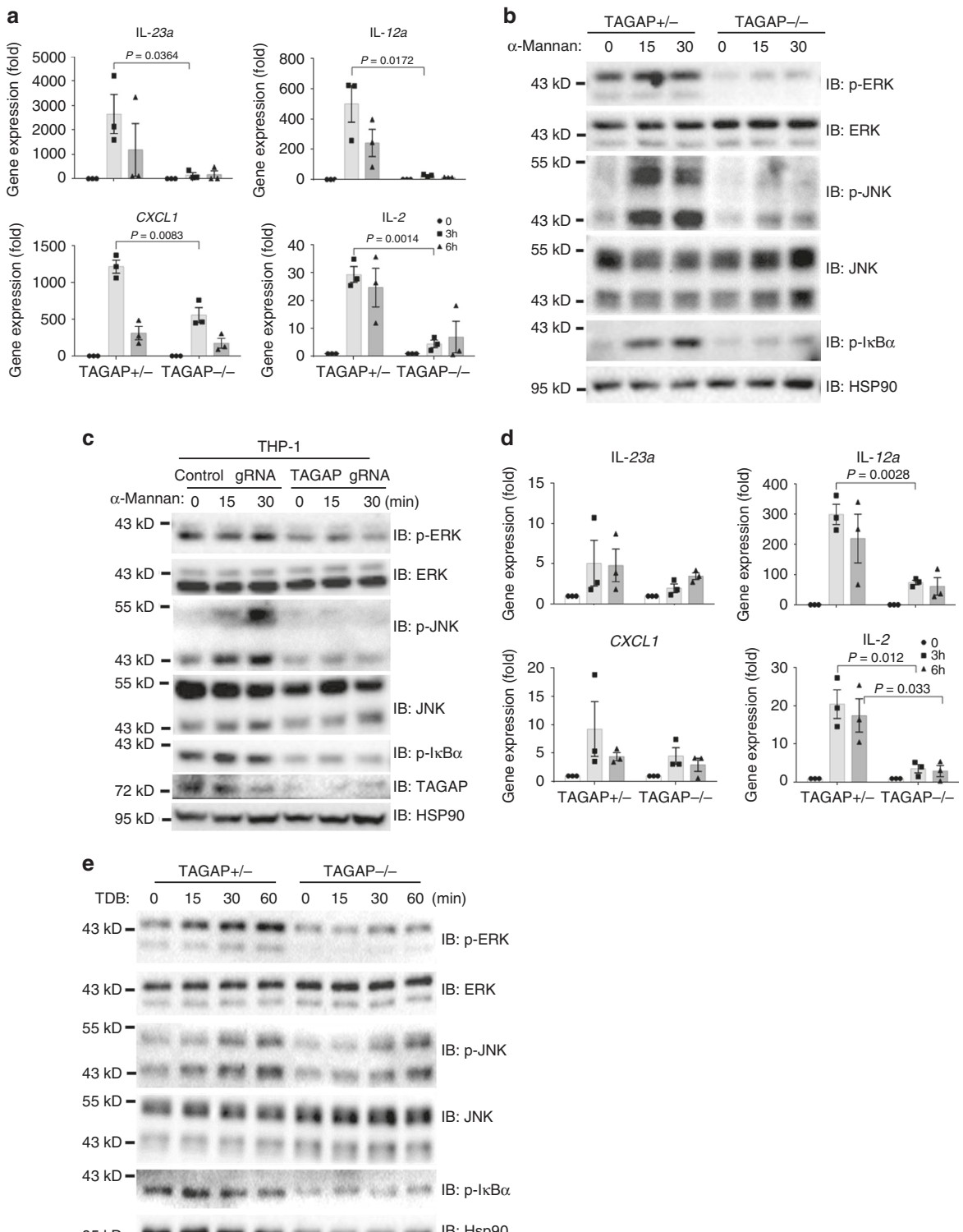

**Fig. 2 TAGAP is required for Dectin-2/3 and mincle ligands-induced signaling activation. a** BMDMs from heterozygous control mice or TAGAP-deficient mice were stimulated with α-Mannan (100 μg/ml) for 3 and 6 h, followed by real-time PCR analysis of indicated genes. **b** BMDMs from heterozygous control mice or TAGAP-deficient mice were stimulated with α-Mannan (100 μg/ml) for indicated times, followed by western blot analysis of indicated protein expression. **c** Control- gRNA or TAGAP-gRNA knocked down THP-1 cells were stimulated with α-mannan (100 μg/ml) for indicated times, followed by western blot analysis of indicated proteins. **d** BMDMs from heterozygous control mice or TAGAP-deficient mice were stimulated with TDB (50 μg/well) for 3 and 6 h, followed by real-time PCR analysis of indicated genes. **e** BMDMs from heterozygous control mice or TAGAP-deficient mice were stimulated with TDB (50 μg/well) for indicated times, followed by western blot analysis of indicated protein expression. $*P < 0.05$; $**P < 0.01$; $***P < 0.001$ based on two-sided unpaired $t$ test **a**, **d**. All error bars represent SEM of technical replicates. Data are representative of three independent experiments.

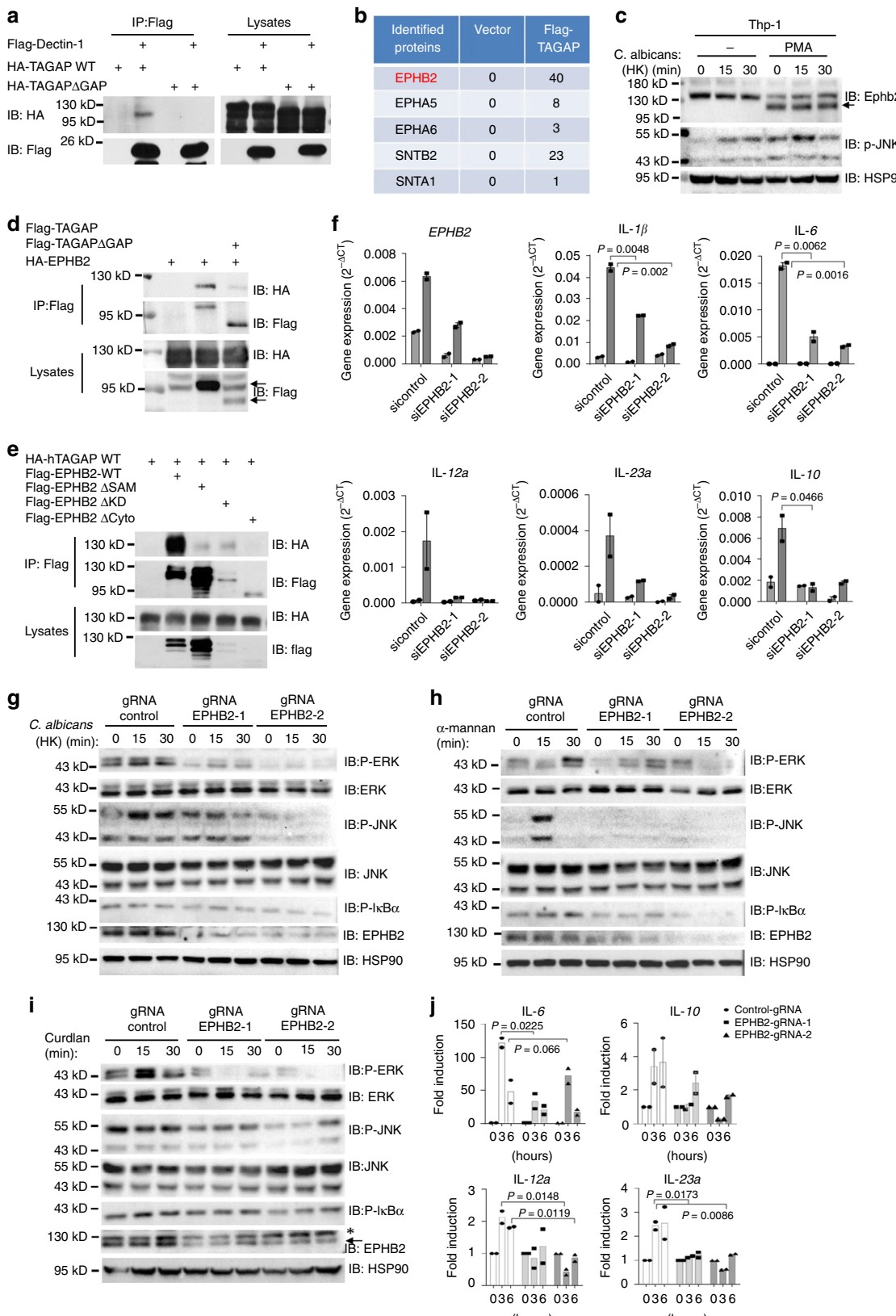

Phosphorylated EPHB2 did not have significant change in TAGAP knocked down THP-1 cells compared with the control cells after heat-killed *C. albicans* stimulation, which suggests that TAGAP functions downstream of EPHB2 in Dectin-1 pathway (Supplementary Fig. 3H). Prediction software indicated that residues Y310, Y596, and Y730 of TAGAP could be phosphorylated by a tyrosine kinase, so we explored whether EPHB2 phosphorylates TAGAP at these three sites. Interestingly, EPHB2 can strongly phosphorylate wild-type TAGAP, whereas the mutation of TAGAP Y310 largely abolished its phosphorylation,

**Fig. 3 TAGAP-binding partner EPHB2 is important for antifungal signaling activation. a** HEK293T cells were transfected with indicated plasmids, and cell lysates were immunoprecipitated with anti-Flag antibody, followed by immunoblot analysis for indicated proteins. **b** Flag-TAGAP stable transfected THP-1 cells were differentiated with PMA (25 ng/mL) for 3 days, followed by immunoprecipitation by Flag antibody. Protein elution was analyzed by mass spectrometry analysis. Graph represents mass spectrometry result of TAGAP-interacting proteins. **c** Human THP-1 cells were non-polarized or polarized by adding PMA (25 ng/mL) for 3 days, and were stimulated with heat-killed *C. albicans* (MOI = 2) for 0, 15, and 30 min, followed by western blot analysis of indicated proteins. **d**, **e** HEK293T cells were transfected with indicated plasmids, and cell lysates were immunoprecipitated with anti-Flag antibody, followed by immunoblot analysis for indicated proteins. **f** Control siRNA or EPHB2 siRNA-transfected BMDMs were stimulated with Curdlan (100 μg/ml) for 3 h, followed by real-time PCR analysis of indicated genes. **g–i** THP-1 cells infected with control gRNA or EPHB2-gRNA were stimulated with heat-killed *C. albicans* (MOI = 2), Curdlan (100 μg/ml) or α-Mannan (100 μg/ml) for the indicated times, followed by western blot analysis of indicated proteins. **j** THP-1 cells infected with control gRNA or EPHB2-gRNA were stimulated with heat-killed *C. albicans* (MOI = 2) for the indicated times, followed by real-time PCR analysis for indicated genes. *P < 0.05; **P < 0.01; ***P < 0.001 based on two-sided unpaired *t* test **f**, **j**. Real-time PCR data of **f** and **j** were collected from two independent experiments. All error bars represent SEM of technical replicates. Data are representative of three independent experiments.

which indicates that EPHB2 mainly phosphorylates TAGAP at site of Y310, although it also phosphorylated Y596 and Y730 of TAGAP to a lesser extent (Fig. 4e, f). TAGAP protein migrated slower on the sodium dodecyl sulfate–polyacrylamide gel electrophoresis (SDS-PAGE) gel when cells were co-transfected with wild-type EPHB2, which is characteristic of protein phosphorylation (Fig. 4e, arrows indicated). Interestingly, endogenous phosphorylation of TAGAP was totally abolished in EPHB2-knocked down THP-1 cells after heat-killed *C. albicans* stimulation (Supplementary Fig. 3I). The phosphorylation of TAGAP Y310 has been identified by another research group[27], and this site is quite conservative among different species (Fig. 4g). Restoration of wild-type TAGAP but not TAGAP Y310F mutant greatly increased heat-killed *C. albicans*-induced ERK and p38 activation in TAGAP-gRNA knocked down THP-1 cells (Fig. 4h). Since CARD9 was reported as an adaptor protein which mediated signaling activation after Dectin-1 ligand stimulation[12], we explored whether TAGAP could bind CARD9. Wild-type TAGAP, but not Y310F mutant of TAGAP can bind CARD9, and this result was further confirmed by immunofluorescence staining (Fig. 4i, j). Interestingly, it seems that TAGAP Y310F lose the membrane localization, and exclusively localized in the nuclear (Fig. 4j). Next, we found that endogenous TAGAP, EPHB2, and CARD9 bound together in a complex after heat-killed *C. albicans* stimulation in THP-1 cells (Fig. 4k). SYK and CARD9 colocalized in the wild-type macrophages after Curdlan and heat-kill *C. albicans* stimulation, whereas the colocalization between SYK and CARD9 was almost abolished in TAGAP-deficient macrophages after heat-kill *C. albicans* stimulation (Fig. 4l–m). Together, these data indicate that EPHB2 has an essential role in the activation of antifungal signaling pathway by phosphorylating TAGAP at site of Y310 for the recruitment of CARD9.

**TAGAP expression level correlates positively with Th17 and Th1 cell abundance in mice**. We explored whether TAGAP deficiency lead to any immune cell developmental defect, and we did not find any adaptive immune cell early development abnormalities, such as T-cell development in the thymus, and CD4$^+$ and CD8$^+$ percentages in the lymph nodes, T-cell activation in the lymph nodes, B-cell subset development including regulatory T cells, recirculating B cells, or mature follicular and marginal zone B cells as well as B1 B cells in TAGAP-deficient mice (Supplementary Fig. 4A–E), which indicates that TAGAP is not involved in earlier T and B-cell development.

Antifungal innate immunity is highly associated with T helper cell differentiation, especially Th17 cell polarization[1,28,29]. We found that Th17 population was markedly reduced in the lymph nodes and spleen of TAGAP-deficient mice compared with control mice, whereas the IFN-γ expressing Th1 population was

marginally reduced in the lymph nodes, and was unchanged in the spleen of TAGAP-deficient mice (Fig. 5a and Supplementary Fig. 5). This was further confirmed by the MOG$_{35–55}$ priming experiment: the difference in Th17 cell polarization between control mice and TAGAP-deficient mice was even more marked after MOG$_{35–55}$ immunization and re-stimulation, and Th1 cell polarization was also greatly impaired in TAGAP-deficient mice compared with control mice (Fig. 5b). For this experiment, mice were primed with MOG$_{35–55}$ plus heat-killed *M. tuberculosis* H37Ra, which is an adjuvant known to activate the Dectin-1 pathway in macrophages[30,31]. These results are consistent with our earlier finding that there is defect in Dectin-1 pathway activation in TAGAP-deficient BMDMs (Fig. 1b–h). Th17 and Th1 cell polarization was decreased in TAGAP-deficient mice primed with MOG$_{35–55}$ plus heat-killed yeast *C. albicans* as an adjuvant, but there was no difference in terms of T helper cell differentiation when primed with MOG$_{35–55}$ plus CFA only (Fig. 5c, d). Consistently, naive T cells from TAGAP-deficient mice did not show any Th17 or Th1 cell polarization defect compared with that from control mice, which indicates that TAGAP did not play an essential role in intrinsic T-cell differentiation (Fig. 5e). It was well-known that *C. albicans* can polarize T-cell differentiation into Th17 cells in vivo, and we incubated CD4$^+$ T cells isolated from *C. albicans*-infected mice with DCs from heterozygous control mice or TAGAP-deficient mice, and re-stimulated the cells with heat-killed *C. albicans*. TAGAP-deficient DCs showed a compromised ability to direct T cells differentiating into Th17 cells, which indicates that TAGAP deficiency in innate immune cells such as macrophages or DCs results in a defect in "directing" Th17 cell polarization (Fig. 5f). Together, these data indicate that TAGAP directs T helper cell differentiation through regulating innate immune signaling pathways.

T helper cells, especially Th17 cells play an essential role for host defense against fungi, and *TAGAP* gene polymorphism was reported susceptible to candidemia[21], so we explored whether TAGAP-deficient mice have any defect in antifungal defense. After *C. albicans* infection, TAGAP-deficient mice died much quicker than heterozygous control mice, and weight loss was also much quicker in TAGAP-deficient mice than that of control mice (Fig. 6a, b). Liver and kidney had significant higher fungal burden in TAGAP-deficient mice than that of control mice (Fig. 6c). Consistent with earlier finding, Th17 cell abundance was significantly reduced in the lymph nodes of TAGAP-deficient mice than that of heterozygous control mice (Fig. 6d). Overall, these data indicate that TAGAP-deficient mice had defect in against fungal infection.

To explore whether TAGAP-deficient DCs have an antigen presenting defect, we incubated CD4$^+$ T cells from 2D2 transgenic mice with DCs which from heterozygous control mice or TAGAP-deficient mice in the presence of MOG$_{35–55}$, and examined Th17 cell differentiation. There was no defect in Th17 cell differentiation under these conditions, which indicates that

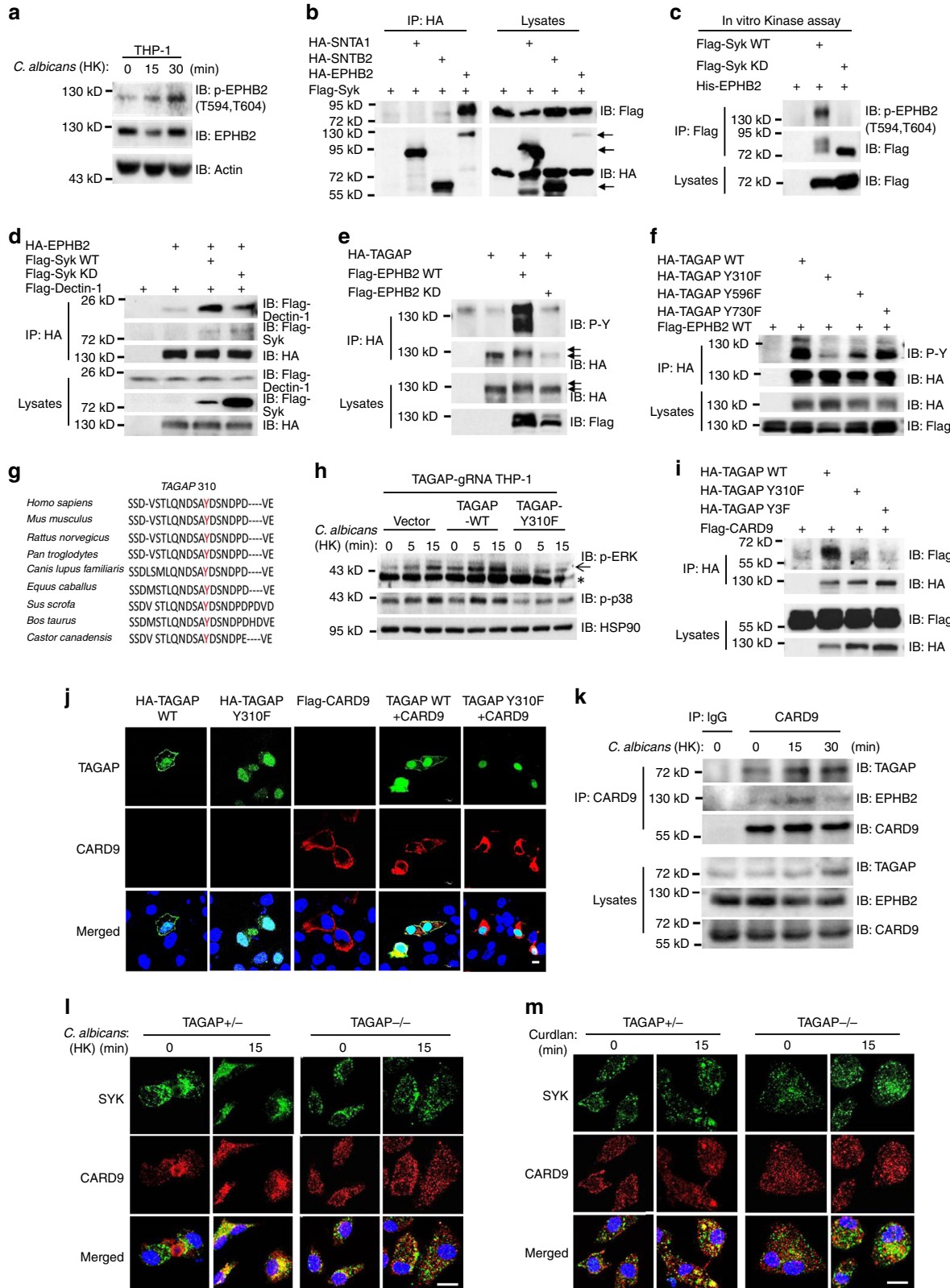

TAGAP-deficient DCs have normal antigen presenting abilities (Supplementary Fig. 6).

**TAGAP expression level correlates positively with Th17 cell abundance in humans.** To explore whether this TAGAP-Th17 correlation exists in human, we collected PBMCs from healthy volunteers and identified the individuals who carried diseases-associated *TAGAP* SNPs, including rs1738074 and rs3127214. Rs1738074 is located in the first exon of *TAGAP*, and having a "C" residue at this locus predisposes carriers to MS[32], RA[33], and IBD susceptibility[32–34]. Rs3127214 is located upstream of *TAGAP*, and having a "T" residue at this locus is associated with susceptibility to candidemia[21]. We found that PBMCs from

**Fig. 4 EPHB2 phosphorylates TAGAP at tyrosine 310. a** THP-1 cells were stimulated with heat-killed *C. albicans* (MOI = 2) for indicated times, followed by western blot analysis for indicated proteins. **b** HEK293T cells were transfected with indicated plasmids, and cell lysates were immunoprecipitated with anti-HA antibody, followed by immunoblot analysis for indicated proteins. **c** HEK293T cells were transfected with Flag-Syk WT and KD, and cell lysates were immunoprecipitated with anti-Flag antibody, followed by in vitro kinase assay by incubating with purified His-EPHB2 as substrate. Immunoblot was conducted for the analysis for indicated proteins. **d–f** HEK293T cells were transfected with indicated plasmids, and cell lysates were immunoprecipitated with anti-HA antibody, followed by immunoblot analysis for indicated proteins. **g** Schematic presentation of conservativeness of TAGAP Y310 site between different species was shown. **h** TAGAP-gRNA-infected THP-1 cells were infected with indicated plasmids by lentivirus expressing vectors, and cells were stimulated with heat-killed *C. albicans* (MOI = 2) for indicated times, followed by western blot analysis of indicated protein expression. **i** HEK293T cells were transfected with indicated plasmids, and cell lysates were immunoprecipitated with anti-HA antibody, followed by immunoblot analysis for indicated proteins. **j** Hela cells were transfected with indicated plasmids, and immunofluorescence was done by staining HA and Flag antibodies. Scale bars = 10 μm. **k** THP-1 cells were stimulated with heat-killed *C. albicans* (MOI = 2) for indicated times, and cell lysates were immunoprecipitated with anti-CARD9 antibody, followed by immunoblot analysis for indicated proteins. **l**, **m** BMDMs from heterozygous control or TAGAP-deficient mice were left untreated or treated with heat-killed *C. albicans* (MOI = 2) **l** or Curdlan (100 μg/ml) **m** for the indicated time, followed by immunofluorescence analysis of colocalization of endogenous SYK and CARD9. Scale bars = 10 μm. Data are representative of three independent experiments.

individuals carries the *TAGAP* rs1738074 C/C genotype had the significant higher *TAGAP* mRNA expression levels compared with that from rs1738074 T/C and T/T carriers, whereas PBMCs of rs3127214 T/T carriers had significantly lower *TAGAP* mRNA levels compared with that of rs3127214 C/T and C/C carriers (Fig. 7a). Furthermore, Th17 cell abundance was positively correlated with the *TAGAP* mRNA levels in PBMCs from both of two genotypes, which was consistent with the finding which we got from mouse study (Fig. 5a–c and Fig. 7a–c). The lower Th17 abundance in the PBMCs of rs3127214 polymorphism carrier could very well explain its susceptibility to candidemia, as Th17 cells were known to play the most critical role in host defense against fungi, such as *C. albicans*[1,2,35,36]. Although the higher Th17 level in the PBMCs of rs1738074 carriers could explain its autoimmune diseases susceptibility, as inflammatory Th17 cells were known to be involved in the pathogenesis of many autoimmune diseases, such as MS and RA[3,28,37]. Next, we stimulated PBMCs from individuals with different *TAGAP* genotypes with Curdlan and α-Mannan. Strikingly, PBMC-induced proinflammatory gene expression in response to different stimuli was positively correlated with *TAGAP* mRNA level (Fig. 7d, e). This data further confirms that human TAGAP has a pivotal role in activation of the antifungal signaling pathway in human cells.

**Broad-spectrum tyrosine kinase inhibitors Dasatinib and Vandetanib attenuate EAE severity.** SNPs in *TAGAP* are associated with multiple autoimmune diseases, including MS, so we explored the role of TAGAP in a mouse model of EAE. TAGAP-deficient mice showed a delay in the onset of neurological impairment, which occurred ~ 14 days after immunization, compared with 11 days for heterozygous control mice (Fig. 8a). TAGAP-deficient mice also showed much lower disease severity, with an average EAE score of 1.6 ± 0.49, compared with an average score of 3.3 ± 0.5 for their heterozygous control mice (P = 0.0007) (Fig. 8a). Consistent with the clinical scores, perivascular leukocyte infiltration and demyelination were much more prominent in the spinal cords of heterozygous control mice compared with TAGAP-deficient mice, as determined by H&E staining and LFB staining (Fig. 8b). There was a marked decrease in the expression levels of 'signature' IL-17-responsive genes, such as *CXCL1*, *CXCL2*, *IL-6*, and *IL-1β*, in the spinal cords of TAGAP-deficient mice compared with heterozygous control mice, as determined by real-time PCR analysis (Fig. 8c), which is consistent with our earlier finding that Th17 cell polarization was defective in TAGAP-deficient mice (Fig. 5a–c). There was a great degree of mononuclear cell infiltration in the brain white matter and spinal cords of heterozygous control mice than in TAGAP-deficient mice (Fig. 8d). One study found that CD4+ T cells from TAGAP-deficient mice had intrinsic defect in Th17

polarization[23], whereas we did not find naive T cells from TAGAP-deficient mice have intrinsic differentiation defect (Fig. 5e). To further explore whether TAGAP plays any role in CD4+ T-cell differentiation, we set up EAE model by using Rag2-deficient mice transferred CD4+ T cells from control or TAGAP-deficient mice. EAE clinical score was comparable between the Rag2-deficient mice transferred with CD4+ T cells from wild-type or TAGAP-deficient mice (Supplementary Fig. 7A). The Th17 and Th1 cells in the peripheral blood and brain were also comparable between Rag2-deficient mice transferred with wild-type or TAGAP-deficient CD4+ T cells (Supplementary Fig. 7B–E). These data strongly indicate that TAGAP did not play an intrinsic role in CD4+ T-cell differentiation.

As EPHB2 can be phosphorylated by Syk, and this phosphorylation can be induced by activation of the Dectin-1 pathway by heat-killed *C. albicans* (Fig. 4a–c), we explored the functional role of EPHB2 kinase activity in the antifungal signaling pathways by using broad-spectrum tyrosine kinase inhibitors Dasatinib and Vandetanib, which were shown to be able to inhibit EPHB2 kinase activity[39], and these two drugs can also inhibit the activity of several other kinases. We first confirmed the inhibition role of Dasatinib and Vandetanib on EPHB2 kinase activity (Fig. 8e, f). Vandetanib and Dasatinib strongly inhibited EPHB2-mediated TAGAP phosphorylation, and significantly inhibited Curdlan- and α-Mannan-induced proinflammatory gene expression (Fig. 8g–i). Consistently, SYK and CARD9 colocalization was abolished in Vandetanib and Dasatinib pretreated THP-1 cells after heat-killed *C. albicans* stimulation (Supplementary Fig. 8). Interestingly, SYK translocated to the cell membrane after heat-killed *C. albicans* stimulation in wild-type THP-1 cells, and this phenomenon was not quite obvious in mouse BMDMs, which may be owing to species difference of the human and mouse cells. These results indicate that the kinase activity of EPHB2 may have an important role in Dectin ligands-induced proinflammatory gene expression. Owing to the lack of specificity of Vandetanib and Dasatinib toward EPHB2, we cannot exclude the possibility that the effect of Vandetanib and Dasatinib may partially owing to inhibition of other kinases. Antifungal signaling pathways activation is believed to have an essential role in Th17 cell differentiation in vivo[35,40], and Vandetanib and Dasatinib can block Dectin ligands-induced proinflammatory cytokine expression, so we investigated whether Vandetanib and Dasatinib can inhibit T helper cell differentiation and EAE severity. Strikingly, EAE score was greatly reduced after two weeks of oral gavage of Vandetanib and Dasatinib. Vandetanib was more effective than Dasatinib in terms of EAE treatment, which may be owing to the stronger inhibition of proinflammatory gene induction by Vandetanib compared with Dasatinib (Fig. 8j). There was

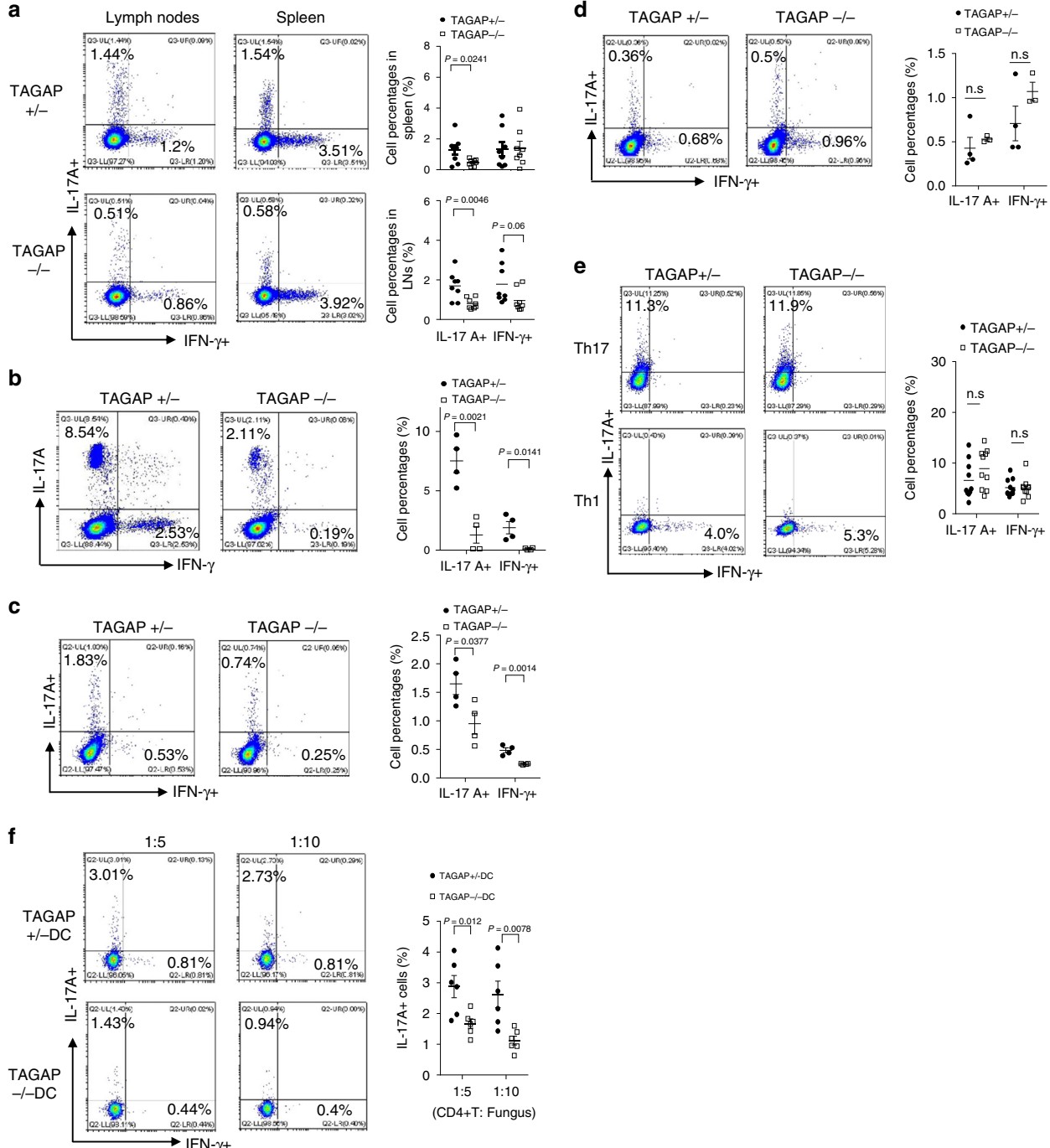

significantly less Th17 cells brain infiltration after Vandetanib and Dasatinib treatment, and Th17 and Th1 cell polarization were both significantly reduced in the spleen after treatment with these inhibitors (Fig. 8k). Overall, these data indicate that EPHB2 is a promising target for EAE treatment, and these two existing drugs could potentially be used to treat Th17-mediated autoimmune diseases/inflammation, including MS.

## Discussion

In this study, we provide evidence that TAGAP is an adaptor in antifungal signaling pathway. TAGAP deficiency in innate immune cells, such as macrophages and DCs, results in a defect in the secretion of proinflammatory cytokines, such as IL-23a and IL-12a, which affects Th17 and Th1 cell differentiation. This

dysregulation of Th17 and Th1 cells increases susceptibility to autoimmune diseases and fungal infectious diseases, including MS and candidemia (Fig. 8l). We also showed that two existing drugs Vandetanib and Dasatinib can block Th17 and Th1 cell polarization in vivo, and had a strong therapeutic effect on mice EAE, which suggests that these two drugs could potentially be used to treat Th17-mediated autoimmune diseases such as MS.

There is an established link between infection and autoimmune diseases in humans, and this link has been attributed to either molecular mimicry between pathogen-derived antigens and self-antigens or chronic activation of the immune response, leading to a breakdown in immunological tolerance and the development of self-antigen-specific T-cell and B-cell responses[29,37]. In recent years, GWAS studies have identified several innate immune genes as being implicated autoimmune diseases; for example, TLR2 is

**Fig. 5 TAGAP is important for T helper cell polarization in mice. a** Cells of lymph nodes and spleens from heterozygous control or TAGAP-deficient mice were analyzed by flow cytometry by gating on CD4+, and analyzed by IL-17A or IFN-γ-producing cells. Right panel indicates quantitative result, and the top right panel represents data from spleen, and the bottom right data represent data from lymph nodes ($n = 8$). **b** Heterozygous control or TAGAP-deficient mice were immunized subcutaneously with 200 μg MOG$_{35-55}$ and 400 μg *Mycobacteria tuberculosis* H37RA in 200 μL of complete Freund's adjuvant (Difco). Lymph nodes were collected 10 days later and single-cell suspensions were prepared. Cells were cultured with MOG$_{35-55}$ (20 μg/mL) for another 3 days, followed by flow cytometry analysis of indicated cell populations. Right panel were quantitative results ($n = 4$). **c** Heterozygous control or TAGAP-deficient mice were immunized as in **b**, except for immunization with 200 μg MOG$_{35-55}$ plus CFA and 400 μg heat-killed *C. albicans* (sc-5314). Right panel were quantitative results ($n = 4$). **d** Heterozygous control or TAGAP-deficient mice were immunized as in **b**, except for immunization with 200 μg MOG$_{35-55}$ plus CFA only ($n = 4$). ns: $P = 0.5226$ and $P = 0.288$. **e** Naive CD4+ T cells were isolated from spleens of heterozygous control or TAGAP-deficient mice, and polarized in vitro for 3 days in the Th17 or Th1 cell polarization conditions. Cells were analyzed by flow cytometry by gating on CD4+, and analyzed by IL-17A or IFN-γ-producing cells. Right panel were quantitative results ($n = 10$). ns: $P = 0.1848$ and $P = 0.9952$. **f** Wild-type mice were injected intravenously with $2 \times 10^5$ live *C. albicans* (sc-5314) in 100 μL PBS. Ten days later, CD4+ T cells were isolated from spleen, and co-cultured with DCs isolated from heterozygous control or TAGAP-deficient mice for another 3 days together with indicated numbers of heat-killed *C. albicans*. Cells were analyzed by flow cytometry by gating on CD4+, and analyzed by IL-17A or IFN-γ-producing cells ($n = 6$). Right panel were qualified results. *$P < 0.05$; **$P < 0.01$; ***$P < 0.001$ based on two-sided unpaired *t* test **a**–**f**. All error bars represent SEM of technical replicates. Data are representative of two independent experiments.

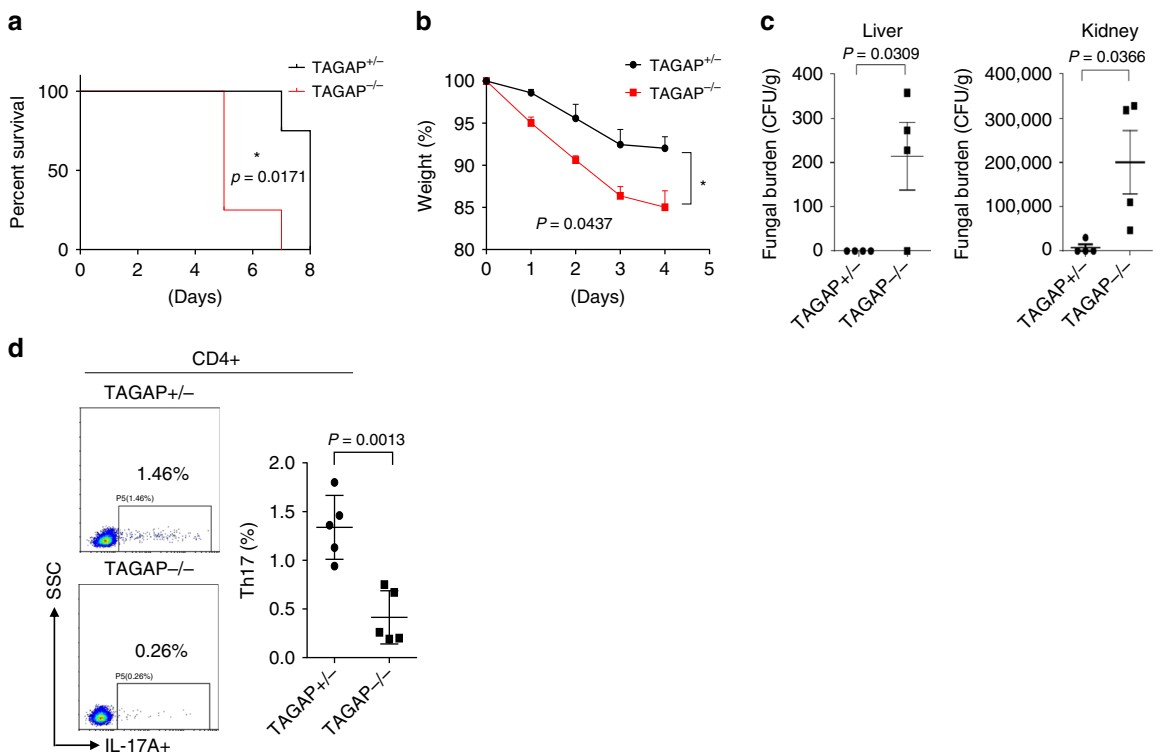

**Fig. 6 TAGAP-deficient mice were susceptible to fungal infection. a**, **b** Heterozygous control mice or TAGAP-deficient mice were infected with live *C. albicans* ($5 \times 10^5$) through tail vein injection, and survival rate **a** or weigh loss **b** was shown ($n = 4$). **c** Heterozygous control or TAGAP-deficient mice were infected with live *C. albicans* ($2 \times 10^5$) through tail vein injection, and after 7 days fungal burden in liver and kidney were assessed as shown ($n = 4$). **d** Heterozygous control or TAGAP-deficient mice were infected with live *C. albicans* ($2 \times 10^5$) through tail vein injection, and after 7 days lymph node cells from heterozygous control or TAGAP-deficient mice were analyzed by flow cytometry by gating on CD4+, and analyzed by IL-17A or IFN-γ-producing cells ($n = 5$). All error bars represent SEM of technical replicates. *$P < 0.05$; **$P < 0.01$; ***$P < 0.001$ based on two-way ANOVA **a**, **b** and two-sided unpaired *t* test **c**, **d**. Data are representative of two independent experiments.

associated with susceptibility to type I diabetes and psoriasis[41]; *CARD9* and *NOD2* to Crohn's disease[42]. These findings provide a genetic link between dysregulation of innate immunity and susceptibility to autoimmune diseases, but the mechanism of dysregulation in innate immune signaling pathways leads to autoimmune diseases remains unclear. In this study, we provide a specific example of innate immunity pathway dysregulation and autoimmune susceptibility. This study not only explains the mechanism of susceptibility of *TAGAP* gene polymorphisms to many autoimmune diseases, but also uncovers an association between antifungal signaling pathways and the autoimmune

diseases susceptibility, which provides novel targets for the treatment of the autoimmune diseases. Further research is needed to clarify the molecular and cellular mechanism of multiple innate immune pathway dysregulation and susceptibility to different autoimmune diseases.

Interestingly, one study found that TAGAP-deficient CD4+ T cells had an intrinsic defect in Th17 polarization, and TAGAP-KO mice were less susceptible to EAE[23]. We got the same phenotype in terms of EAE severity of TAGAP-KO mice, however we did not find the in vitro Th17 differentiation defect in TAGAP-deficient CD4+ T cells, and this was further confirmed by EAE

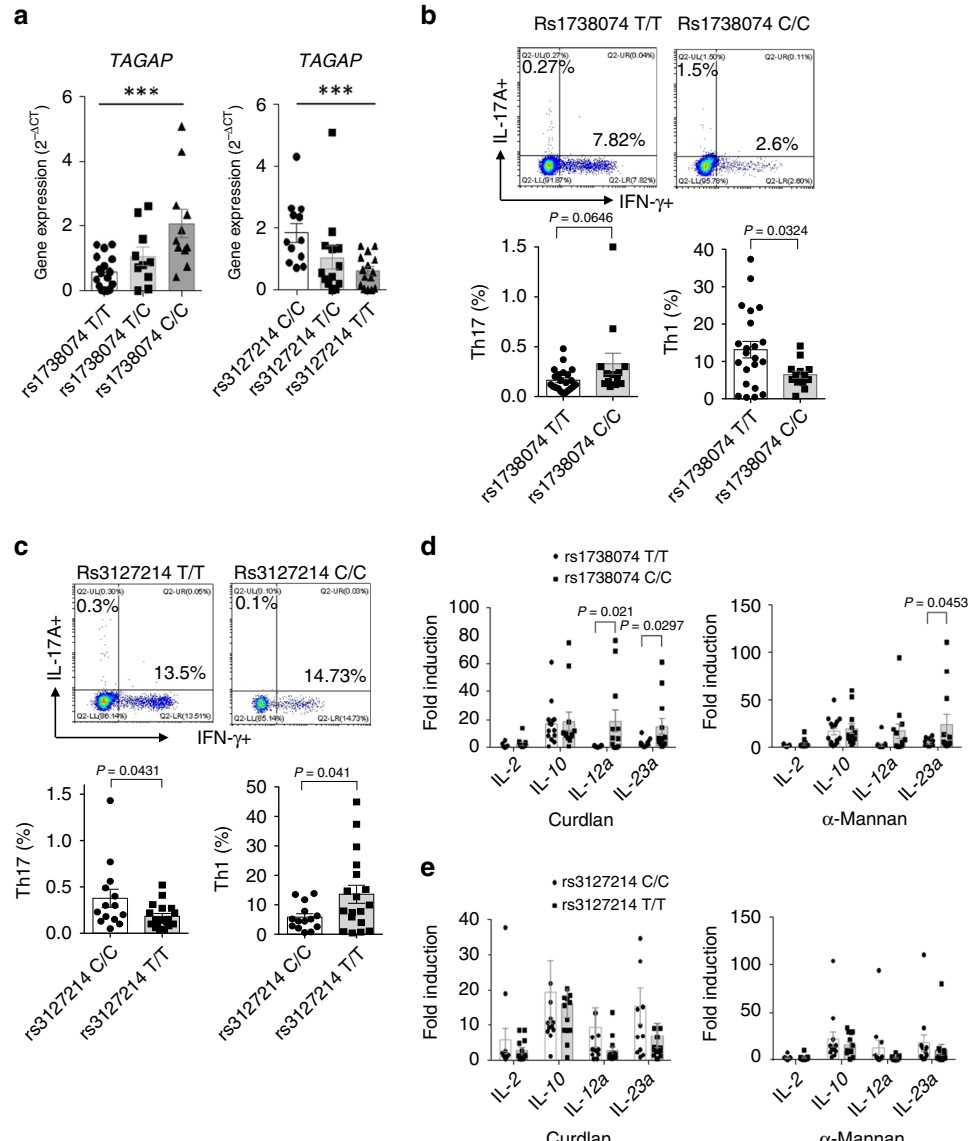

**Fig. 7 TAGAP expression level correlates positively with Th17 cell abundance in humans. a** Human PBMCs were divided in different groups based on the *TAGAP* SNP genotypes, and *TAGAP* mRNA expression was analyzed by RT and real-time PCR. Each dot in the data represent the data from each individual (*n* = 17, 10, 11 for rs1738074; *n* = 12, 13, 15 rs3127214). **b, c** Human PBMCs of *TAGAP* rs1738074 T/T and C/C **b** or *TAGAP* rs3127214 C/C and T/T **c** were analyzed by flow cytometry by gating on CD3+, and analyzed by IL-17A or IFN-γ-producing cells. Lower panel were quantitative results. **d, e** PBMCs of *TAGAP* different genotypes were stimulated with Curdlan (100 μg/ml) or α-Mannan (100 μg/ml) for 3 h, followed by real-time PCR analysis of indicated genes. \*P < 0.05; \*\*P < 0.01; \*\*\*P < 0.001 based on two-sided unpaired *t* test **a**–**e**. All error bars represent SEM of technical replicates.

model on Rag2-deficient mice transferred with CD4+ T cells from control or TAGAP-deficient mice (Fig. 5e and Supplementary Fig. 7). One of possible reason for this discrepancy could be owing to the different T-cell polarization methods used between us, since we noticed that the other group polarized Th17 cells in vitro by adding TGFβ, IL-6, IL-1β, and IL-23a in the presence of anti-CD3, anti-CD28, anti-IFN-γ, and anti-IL-4, whereas we did not add IL-1β and IL-23a for Th17 cell in vitro polarization, as many studies have found that IL-6 plus TGFβ were sufficient for naive CD4+ cell polarization to Th17 cells in vitro[38,43,44]. However, studies found that IL-23a did not further expand Th17 cells in an in vitro polarization condition, instead induce granulocyte-macrophage colony-stimulating factor (GM-CSF) production in existing Th17 cells[45,46]. Although there may be different ways of polarization of Th17 cells in vitro, our EAE model on Rag2-deficient mice clearly indicates that TAGAP-

deficient CD4+ T cells did not have any intrinsic defect (Supplementary Fig. 7).

Eph receptors constitute the largest subgroup of tyrosine kinase receptors, and Eph receptors and ephrin signaling has well-known roles in controlling cell migration and axon guidance. EPHB2 is also involved in cancer initiation and progression, and EPHB2 mutations are present in clinical prostate cancer samples[47–49]. There have been a few reports of the role of Ephrin-B-EPHB2 signaling in immune cells, such as T-cell activation[50]. Our study elucidates the important role of EPHB2 in the host innate immune response to pathogens, such as fungi. We believe that the role of EPHB2 in antifungal signaling is very important, and it will provide a novel target for the treatment of fungal infection and autoimmune diseases, as we have provided evidence that two existing drugs Vandetanib and Dasatinib can inhibit T helper cell differentiation in vivo and diminish mice EAE severity.

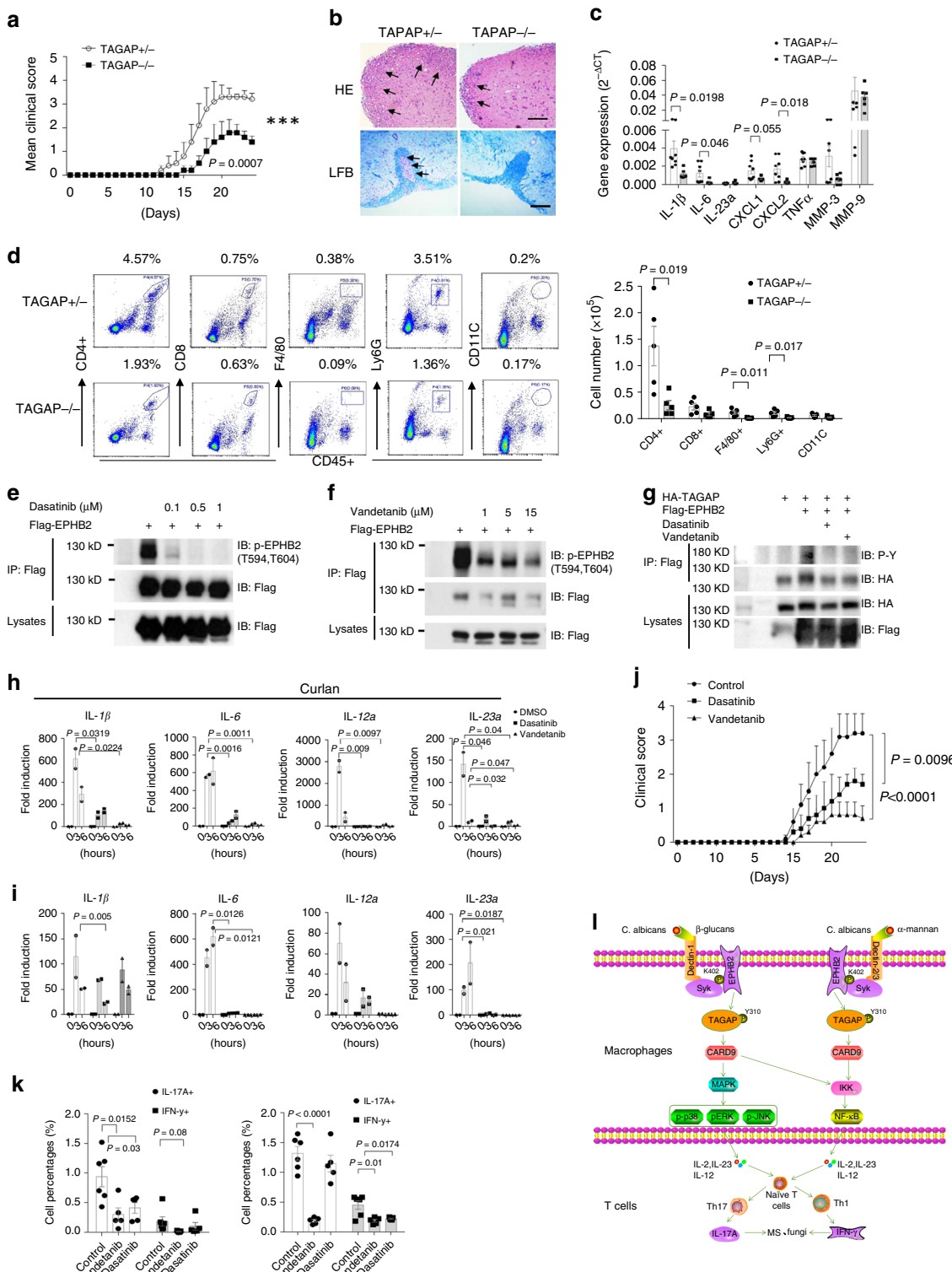

Interestingly, one group has also found that dasatinib had beneficial effects on EAE by lower incidence, attenuation in the severity and a delay in the onset of disease[51]. Although, owing to the lack of specificity of Vandetanib and Dasatinib towards EPHB2, we cannot exclude the possibility that the effect of Vandetanib and Dasatinib may partially owing to inhibition of other kinases. A recent study found that a human *EPHB2* variant (missense p.R745C) affects platelet function, which is owing to partially impaired kinase activity of EPHB2[52]. Interestingly, Syk,

PLC-γ2 and FcRγ phosphorylation were drastically impaired in the patients' cells in response to glycoprotein VI platelet signaling[52]. Notably, Syk, PLC-γ2, and FcRγ also play an important role in antifungal signaling pathway, which suggests a conserved function of EPHB2 in different signaling pathways. A previous study found that EPHA2 functions as a β-glucan receptor for fungal recognition in epithelial cells[53], and further research is needed to explore whether EPHB2 can also function as a receptor or co-receptor to directly recognize fungal components.

**Fig. 8 Vandetanib and Dasatinib attenuate EAE severity. a** EAE Clinical score of heterozygous control mice or TAGAP-deficient mice in EAE model was shown ($n = 5$). **b** Hematoxylin and Luxol fast blue staining of transversal sections of lumbar spinal cords from heterozygous control or TAGAP-deficient mice with EAE. Scale bars represent 200 μm. Arrows in the upper panel indicate inflammatory cells infiltration, and arrows in the lower panel indicate demyelination area. **c** RT and real-time PCR analysis of inflammatory gene expression in spinal cords from EAE mice ($n = 5$) of indicated genotypes were shown. **d** Infiltrating cells in the brains of heterozygous control or TAGAP-deficient mice with EAE ($n = 5$) were isolated at the peak of the disease (21 days after EAE induction), followed by flow cytometry analysis ($n = 5$). **e–g** HEK293T cells were transfected with indicated plasmids, and 24 h after transfection, cells were treated with Dasatinib **e** and Vandetanib **f** at indicated doses for another 24 h, and cell lysates were immunoprecipitated with anti-Flag **e**, **f** or HA **g** antibody, followed by immunoblot analysis for indicated proteins. **h**, **i** BMDMs from wild-type mice were pretreated with Dasatinib (0.3 μM) or Vandetanib (2 μM) for 24 h, and cells were stimulated with Curdlan (100 μg/ml) **h** or α-Mannan (100 μg/ml) **i** for indicated times, followed by RT and real-time PCR analysis of indicated gene expression. **j** Clinical score of control and Vandetanib and Dasatinib-treated mice ($n = 5$) in EAE model was shown. **k** DMSO, Vandetanib or Dasatinib-treated mice were induced EAE, and at the peak of the disease, infiltrating cells in the brains (left panel) or in the spleen (right panel) were analyzed by flow cytometry by gating on CD4$^+$, and analyzed by IL-17A or IFN-γ-producing cells. **l** Schematic presentation of a proposed model was shown. *$P < 0.05$; **$P < 0.01$; ***$P < 0.001$ based on two-way ANOVA **a**, **j** and two-sided unpaired $t$ test **c**, **d**, **h**, **i**, **k**. Real-time PCR data of **h** and **i** were collected from two independent experiments. All error bars represent SEM of technical replicates. Data are representative of three independent experiments.

## Methods

**Mice**. The *Tagap* gene knockout mouse was a kind gift from Bernhard G Herrmann at Max Planck Institute for Molecular Genetics, Germany, and was reported in the original publication by Bauer et al.[22]. The accession number for the gene targeted in this knockout model is NM_145968, which corresponds to the *Tagap* gene. However, Bauer et al.[22] refer to this gene as *Tagap1*. To clarify the gene targeted in these mice, we developed a quantitative reverse transcriptase-PCR method and showed that the targeted mice lacked *Tagap* mRNA, consistent with the accession number referenced in the original publication. See also MGI ID 3615484 for gene information and MGI ID 3603008 for mouse strain information. All of the mice used in this study were housed in the SPF condition. Experimental protocols were approved by the Institutional Animal Care and Use Committee of Tongji Medical College, Huazhong University of Science & Technology. This study was complied with all relevant ethical regulations for animal research.

**Human PBMC study**. Whole blood samples were drawn from each study participant. Genomic DNA was isolated using the Gentra Puregene blood (QIAGEN). All DNA samples were quantified using NanoDrop 2000 (Thermo Scientific, Wilmington, DE, USA) and inspected for quality by agarose gel electrophoresis. Human PBMCs were isolated by Ficoll-paque PREMIUM (17-5442-02, GE Healthcare) from freshly drawn peripheral venous blood from healthy controls according to manufactory instruction. In brief, add Ficoll-Paque media to the centrifuge tube, and layer the diluted blood sample onto the Ficoll-Paque media solution. Centrifuge at $400 \times g$ for 30 to 40 min at 18 °C to 20 °C with brake turned off. Draw off the upper layer containing plasma and platelets using a sterile pipette, leaving the mononuclear cell layer undisturbed at the interface. Wash the layer of mononuclear cells with 1× PBS. Isolated PBMCs were frozen in −80 °C before use. For the stimulation of human PBMCs, previously frozen PBMC from healthy subjects were thawed and resuspended at $1 \times 10^6$ cells/ml in Rosewell Park Memorial Institute (RPMI) 1640 with 10% fetal bovine serum (FBS). Cells were cultured for 1 h at 37 °C before stimulated with Curdlan (100 μg/ml) and α-mannan (100 μg/ml). Cells were then harvested for gene expression and immunoblotting analysis. This study followed the guidelines set forth by the Declaration of Helsinki, and the protocol passed the review of the Ethics Committee of Tongji Medical College, Huazhong University of Science and Technology. All study participants have signed a written informed consent form.

**Mouse BMDMs and BMDCs preparation**. BMDMs were obtained by differentiating bone marrow progenitors from the tibia and femur of 6–8-week-old male or female mice in Iscove's Modified Dulbecco's Media containing 20 ng/ml of M-CSF (Peprotech), 10% heat-inactivated FBS (Invitrogen), 1 mM sodium pyruvate, 100 U/ml penicillin, and 100 μg/ml streptomycin (Invitrogen) for 5–7 days. Cells were then re-plated in 6-well or 12-well plates 1 day before experiments. BMDCs were obtained by differentiating bone marrow progenitors from the tibia and femur of 6–8-week-old male or female mice in RPMI 1640 Media containing 20 ng/ml of GM-CSF (Peprotech) and 10 ng/ml IL-4, 10% heat-inactivated fetal bovine serum (FBS, Invitrogen), 1 mM sodium pyruvate, 100 U/ml penicillin, and 100 μg/ml streptomycin (Invitrogen) for 9 days. BMDCs were then re-plated in 6-well or 12-well plates 1 day before experiments.

**Reagents**. Antibodies of anti-p-IκBα (14D4), anti-p-p65(93H1), anti-p-p38 (D3F9), anti-p-JNK(81E11), anti-p-Raf-1(56A6), anti-p-SYK(C87C1), anti-HA (C29F4), anti-Flag(D6W5B) were bought from Cell Signaling Technology (cat no. 2859, 3033, 4511, 4668, 9427, 2710, 3724, 14793). Antibodies of anti-HA(H9658) and anti-Flag(F1804) antibodies for immunoprecipitation were bought from Sigma (cat no. H9658 and F1804). Antibodies of anti-p-ERK(12D4), anti-p-Tyrosine (PY20), anti-EPHB2(2D12C6), anti-HSP90(AC-16), anti-Actin(C-2), anti-CARD9 (A-8) and anti-GAPDH(2E3-2E10) were bought from SANT CRUZ

BIOTECHNOLOGY (cat no. sc-81492, sc-508, sc-130068, sc-101494, sc-8432, sc-374569 and sc-293335). Antibody of anti-Syk antibody was bought from Abclonal (cat no. A2123). Antibody of anti-p-EPHB2 (T594, T604) was bought from Thermo Fisher (cat no. PA5-38480). Antibody of TAGAP[EPR15593] was bought from Abcam (cat no. ab187664). Antibodies of anti-CD4(GK1.5), anti-CD8(53-6.7), anti-F4/80(BM8), anti-Ly6G(1A8), anti-B220(RA3-6B2), anti-CD11C(N418), anti-IL-17A(TC11-18H10.1), anti-CD3(17A2), anti-CD44(IM7), anti-CD62L (MEL-14), anti-CD21(7E9), anti-CD23(B3B4), anti-IgM(RMM-1) and anti-IgD (11-26 c.2a) were bought from Biolengend (cat no. 100406, 100758, 123116, 127606, 103212, 117324, 506908, 100204, 103010, 104412, 123418, 101608, 406506, 405736). Antibody of anti-IFN-γ(XMG1.2) was bought from eBioscience (cat no. 11-7311-82). Zymosan, D-zymosan and Curdlan were bought from Invivogen (cat no. tlrl-zyn, tlrl-zyd and tlrl-cud). α-mannan was bought from Sigma (cat no. M3640). Luxol Fast Blue MBS Solution (26681) was purchased from Electron Microscopy Sciences. Mycobacteria tuberculosis H37RA and complete Freund's adjuvant were bought from Difco. Bordetella pertussis toxin was bought from Sigma. Naive CD4$^+$ T-cell isolation kit was bought from Miltenyi Biotec (130-104-453). SiRNA against varies genes were bought from GenePharma company. GenMute or PepMute siRNA transfection reagents were bought from SignaGen Company. Dasatinib and Vandetanib were bought from MCE (cat no. HY-10181 and HY-10260B). MOG$_{35-55}$ was bought from R&D systems (2568). Anti-HSP90 (AC-16, SANT CRUZ BIOTECHNOLOGY) used for western blot was diluted as 1:3000, and other antibodies used for western blot were diluted as 1:1000. All of the antibodies for flow cytometry were diluted as 1:200.

**Immunoblot and immunoprecipitation**. Cell were harvested and lysed on ice in lysis buffer containing 0.5% Triton X-100, 20 mM Hepes pH 7.4, 150 mM NaCl, 12.5 mM β-glycerophosphate, 1.5 mM MgCl2, 10 mM NaF, 2 mM dithiothreitol, 1 mM sodium orthovanadate, 2 mM EGTA, 20 mM aprotinin, and 1 mM phenylmethylsulfonyl fluoride for 30 min, followed by centrifuging at 12,000 rpm for 15 min to extract clear lysates. For immunoprecipitation, cell lysates were incubated with 1 μg of antibody at 4° overnight, followed by incubation with A-sepharose or G-sepharose beads for 2 h, and the beads were washed four times with lysis buffer and the precipitates were eluted with 2× sample buffer. Elutes and whole cell extracts were resolved on SDS-PAGE followed by immunoblotting with antibodies. Densitometric quantification of western blot was performed on images of scanned films using Image J software.

**Tyrosine phosphorylation detection**. For tyrosine phosphorylation experiments, cells were transfected as indicated, followed by lysing with a 1% SDS solution. The lysates were then sonicated for 15 s on ice to disrupt the DNA. The lysates were boiled at 95 °C for 10 min to dissociate the protein interaction. The boiled samples were diluted with co-IP buffer to 0.1% SDS and then centrifuged at 12,000 rpm for 10 min, after which the pellet was discarded. Cell lysates were incubated with 1 μg of antibody at 4° overnight, followed by incubation with A-sepharose or G-sepharose beads for 2 h, and the beads were washed four times with lysis buffer and the precipitates were eluted with 2× sample buffer. The precipitates were resolved by SDS-PAGE and subjected to western blotting analysis by using anti-p-Tyrosine antibody.

**Lentivirus-medicated gene knockout/knockdown in human cell lines**. pLentiCRISPR-GFP vector was used for CRISPR/Cas9-mediated gene knockout in THP-1 or 293 T-cell lines. In brief, lentivirus vector expressing gRNA was transfected together with package vectors into HEK293T (ATCC) package cells. 48 and 72 h after transfection, virus supernatants were harvested and filtrated with 0.2 μm filter. Target cells were infected twice and sorted by flow cytometry-medicated cell sorting. For some experiments, single cell was plated into 96-well plate by flow cytometry for single clone isolation. Isolated single clones were verified by western

blot and DNA sequencing. In some cases, pool of GFP-sorted cells was used in the experiments.

**Real-time PCR.** Total RNA was extracted from spinal cord with TRIzol (Invitrogen) according to the manufacturer's instructions. Ten microgram total RNA for each sample was reverse transcribed using the SuperScript II Reverse Transcriptase from Thermo Fisher Scientific. The resulting complementary DNA was analyzed by real-time PCR using SYBR Green Real-Time PCR Master Mix. All gene expression results were expressed as arbitrary units relative to expression *Actb* or *GAPDH*.

**Real-time PCR primers sequence.** The real-time PCR primers sequence was shown in the Supplementary table 1.

**RNA-seq.** For the RNA-seq experiment, BMDMs were isolated from heterozygous control or TAGAP-KO mice (three mice each genotype, and cells from each mouse were cultured separately as replicates, and cells from each mouse were divided to two parts as untreated or Curdlan-treated when stimulating with Curdlan). Cells were left untreated or treated with Curdlan (10 μg/ml) for 3 h, and were harvested by Trizol. Samples were sent to the BGI company for RNA-seq analysis. For the data analysis, gene expression from Curdlan-treated control cells/ gene expression from Curdlan-treated TAGAP-KO was calculated as the fold (the gene expression of KO cell was normalized as 1).

**Mass spectrometry identification.** Empty vector or Flag-h TAGAP stable transfected THP-1 cells were differentiated with PMA (25 ng/mL) for 3 days, followed by immunoprecipitation by Flag antibody. Protein was eluted and analyzed by mass spectrometry. Samples were reduced and alkylated in dithiothreitol and iodoacetamide followed by trypsin digestion overnight. Digested samples were injected onto Agilent Zorbax 300SB-C18 0.075 mm × 150 mm column on Eskigent nanoLC system coupled with Thermo LTQ-ETD-Orbitrap. Advion Triversa nanomate served as the nano-ion spray source. MSMS data were searched against Refseq human protein database by Sorcerer Sequest. The searched data set was processed by TPP (Trans-Proteomics Pipeline) and filtered with Peptide Prophet.

**In vitro naive CD4$^+$ T-cell differentiation.** For Th17 and Th1 cell differentiation, naive CD4$^+$CD44$_{low}$CD62L$_{high}$ T cells from heterozygous control mice or TAGAP-deficient mice were isolated by naive T-cell isolation kit and activated with plate-bound 1 μg/mL anti-CD3 and 1 μg/mL anti-CD28 and cultured in the presence of 2 ng/mL TGF-β (Peprotech), 20 ng/mL IL-6 (Peprotech), 5 μg/mL anti-IL-4 (11B11), and 5 μg/mL anti-IFN-γ (XMG1.2) for 3–5 days. For Th1 cell differentiation, isolated naive (CD4$^+$CD44$_{low}$CD62L$_{high}$) CD4$^+$ T cells were cultured on an anti-CD3/CD28 coated plate in the presence of 20 ng/mL IL-12a (Peprotech) and 5 μg/mL anti-IL-4 for 3 days, followed by flow cytometry analysis.

**In vivo CD4$^+$ T-cell priming with MOG$_{35-55}$.** For MOG$_{35-55}$-induced CD4$^+$ T-cell priming, heterozygous control mice or TAGAP-deficient mice were immunized subcutaneously in the abdominal flank on day 0 with MOG$_{35-55}$ and Mycobacteria tuberculosis H37RA in 200 μL of an emulsion of equal volumes of water and complete Freund's adjuvant. Ten days after immunization, draining lymph node cells were prepared, and were cultured for 3 days with MOG$_{35-55}$. For CD4$^+$ T-cell priming with *C. albicans*, wild-type mice intravenously injected with live *C. albicans* ($1 × 10^5$), and 7 days later, CD4$^+$ T cells were isolated by CD4$^+$ T-cell isolation kit from spleen (Miltenyi Biotec), and incubated with heat-killed *C. albicans* (MOI = 1) for another three days, followed by flow cytometry analysis.

**C. albicans infection.** Live *C. albicans* strain SC-5314 ($2 × 10^5$ or $5 × 10^5$ yeast cells in 0.1 ml of 1 × PBS buffer) were injected intravenously into 6–8-week-old littermates of distinct genotypes. Infected mice were monitored daily for weight loss and survival. Fungal burden was measured 5 days after infection. After the livers and kidneys were collected, tissue homogenates were serially diluted and plated on yeast extract–peptone–dextrose agar. Fungal colony-forming units were counted after 24 h after plating.

**Induction and assessment of EAE.** EAE was induced in 8-week-old female heterozygous control mice or TAGAP-deficient mice. Mice were immunized subcutaneously in the abdominal flank on day 0 with MOG$_{35-55}$ and Mycobacteria tuberculosis H37RA (Difco) in 200 μL of an emulsion of equal volumes of water and complete Freund's adjuvant (Difco). On days 0, 3, and 7, each mouse was injected intraperitoneally with 0.2 μg of purified Bordetella pertussis toxin. Mice were weighed and were assigned scores daily for neurological signs according to the following scale: 0, no disease; 1, decreased tail tone or slightly clumsy gait; 2, tail atony and/or moderately clumsy gait and/or poor righting ability; 3, limb weakness; 4, limb paralysis; 5, moribund state or death. All experiments were done in a 'blinded' way, which was established and maintained throughout the course of experiments. For evaluation the effect of dasatinib and vandetanib on EAE phenotype, mice were immunized with MOG$_{35-55}$ at day 0, and were randomly

divided into vehicle group (dimethyl sulfoxide (DMSO) 1% in water, 100 μL, by oral gavage) or dasatinib group and vandetanib group, which was administered in the doses of 20 mg/kg body weight/day for 2 weeks started at day 1 after MOG$_{35-55}$ immunization (diluted in 1% DMSO, 100 μL by oral gavage). For the setting up of EAE model on Rag2$^{-/-}$ mice, the Rag2$^{-/-}$ mice were intravenously injected by flow cytometry-sorted CD4$^+$ T cells from wild-type littermate control mice or TAGAP-deficient mice ($3 × 10^6$/mouse), followed by the immunization by MOG$_{35-55}$ as indicated above. All mice were housed in the Biological Resources Unit of the Huazhong University of Science & Technology, and all mouse experimental protocols were approved by the Institutional Animal Care and Use Committee of the Tongji Medical College, Huazhong University of Science & Technology with the Public Health Service policy on humane care and use of laboratory animals.

**Isolation and analysis of CNS inflammatory cells.** Brains were homogenized in ice cold tissue grinders, filtered through a 100 μm cell strainer and the cells collected by centrifugation at 400 *g* for 5 min at 4 °C. Cells were resuspended in 10 ml of 30% Percoll (Amersham Bioscience) and centrifuge onto a 70% Percoll cushion in 15-ml tubes at 800 *g* for 30 min. Cells at the 30–70% interface were collected and were subjected to flow cytometry.

**Statistics.** Non-parametric statistics was applied to compare differences between two groups. Unpaired two-sided *t* test was used to derive all of the *P* value, except for the clinical scores and weight change curve, which were analyzed with two-way ANOVA for multiple comparisons. $P < 0.05$ was considered to be significant. Results are shown as mean and the error bar represents SEM technical or biological replicates as indicated in the figure legend.

**Reporting summary.** Further information on research design is available in the Nature Research Reporting Summary linked to this article.

## Data availability
The source data underlying Figs. 1a, c, e, 2a, d, 3f, j, 5a–f, 6a–d, 7a–e, 8a, c, d, h–k, Supplementary Fig. 1b, d, 2a–f, 4a–e, 6a, b, 7a, c, E, and 9a–g are provided as a Source Data file. The RNA-SEQ data have been deposited into the GEO database, with the accession number of GSE146156. Other data that support the study are available from the corresponding author upon reasonable request.

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

## Acknowledgements

This investigation was supported by the Independent innovation grant from Huazhong University of Science & Technology, and the "Youth Science 'Chenguang project' from Wuhan Science and Technology Bureau (grant number: 2017050304010294), and the grant from the National Natural Science Foundation of China (grants 81871280), and the Junior Thousand Talents Program of China (to C.H.W.).

## Author contributions

J.C., R.H., W.S., R.G., and Q.P. did the experiments; L.Z., Y.D., X.M., X.G., and H.Z. contributed to the experiments; C.T., J.W., W.Z., and Q.K.W. helped to get human PBMC samples; H.B. provided the TAGAP-deficient mice; X.W. provided the 2D2 transgenic mice; J.M. and X.L. provided reagents and participated in discussion; Q.K.W. and X.L. contributed reagents, and participated in discussion; B.N.M. helped to polish the English writing, and participated in discussion; L.Z. and C-J.Z. provided reagents, and independently repeat the experiment of Fig. S7; J.C., R.H., and C.W. analyzed the data; C.W. wrote the manuscript, and oversaw the experiments with C-J.Z. and W.S.

## Competing interests

The authors declare no competing interests.
