## [Peer Review File · Nature Communications]

Reviewers' comments:

Reviewer #1, expert in Th17 (Remarks to the Author):

Chen et al. demonstrated that TAGAP and EPHB2 were critical in dectin ligands-induced production of proinflammatory cytokines and its signal transduction in macrophages. EPHB2 associates and phosphorylates TAGAP, then phospho-TAGAP can bind to CARD9 to activate MAPK and NFκB pathways. TAGAP deficient mice showed ameliorated EAE symptom, and the authors concluded that reduced Th17 responses in TAGAP deficient mice was due to the incomplete production of dectin-dependent cytokines from macrophages. Although discovery of the function of TAGAP and EPHB2 in dectin-signaling is provocative, several critical issues are of concern. The authors effort to eliminate these concerns is of great importance.

Major concerns

1. Impaired EAE symptom in TAGAP-deficient mice has previously reported (Tamehiro et al, Immunol Cell Biol., 95: 729, 2017), showing that T cell intrinsic defect of Th17 differentiation. Indeed, effect of TAGAP deficiency in Th17 differentiation from naive T cells in vitro (Fig. 5e) was completely opposite from the reported results. The authors should quote the paper and reconcile or discuss about the discrepancy. Because obvious contrary indication exists, the author should show more persuasive results in order to insist that reduction of Th17 induction in vivo was due to the dectin/EPHB2/TAGAP-dependent, impaired cytokine production from macrophages.

for example:

Transfer of wild type macrophages into TAGAP-deficient mice can restore reduced EAE?
Dectin (or EPHB2) deficient mice show impaired EAE or Th17 induction?

Minor points

2. In Fig.5, there are several mistakes? in the number of % in quadrant. The % written are in some cases different from the one written in raw data. (5a left top 0.74%>1.44%, also in 5e and 5f). This kind of mistake tends to give negative impression about reliability of the author's data management.

Comments

3. Since involvement of TAGAP and EPHB2 in dectin-signaling was firstly shown in the manuscript, one would be more interested in knowing if TAGAP-deficient mice would be sensitive for fungal/yeast infection such as *Pneumocystis carinii* or *Candida albicans*, which was seen in dectin1/2-deficient mice.

Reviewer #2, expert in antifungal innate immunity (Remarks to the Author):

This manuscript describes the role of TAGAP and EphB2 in macrophage activation, mainly in response to heat-killed *C. albicans*. It contains a very large amount of data, much of which are convincing and support some of the conclusions. Specific comments:

- 1) While the manuscript is easy to comprehend, its English could be improved by a thorough editing to correct the numerous minor grammatical errors. It would also make the manuscript significantly easier to review if the lines were numbered and if the figure legends were located near the figures.
- 2) In the Western blots that show the phosphorylation of specific proteins, such as in Figures 1b, d, f, 2 b and d, the total amount of unphosphorylated proteins should also be shown. For example, in addition to probing the blots for p-ERK, the amount of total ERK should also be shown. Also, in Fig 1d, the macrophages in the zymosan experiments appear to be constitutively activated because ERK is highly phosphorylated even at time 0. It would also strengthen the results if densitometry data from replicate Western blots were shown in the supplementary data.
- 3) In Fig. 1 e and 2a and c, the real-time PCR data from all 3 experiments should be combined rather than showing the results of single experiment.
- 4) Why was heat-killed *C. albicans* used the stimulation experiments rather than live organisms?
- 5) RNA-seq data should be deposited in a public database.
- 6) The Western blotting data showing the differences in phosphorylation of IKbalpha and ERK in Fig. 2 are not very convincing. Again, total IKbalpha and ERK should be shown and densitometry results of replicate Western blot should be presented.
- 7) The description of the mass spectrometry experiments whose results are shown in Fig. 3b is so brief that the reader cannot understand what was actually done. What cells were used and what protein was the target of the immunoprecipitation? Was it TAGAP? If so, were any Dectins detected?
- 8) It would strength the IP data if it could be shown that TAGAP interacts with Dectin-1 and EphB2 in either mouse or human macrophages rather than in transfected HEK cells. Also, the authors should determine whether knockdown of EphB2 blocks the association of Dectin-1 and TAGAP, as suggested by their model. If antibodies are not available for performing immunoprecipitation of untagged proteins, then the authors could at least perform proximity ligation assays in intact cells. In addition, it would be useful to know if Dectin-2 and 3 acted similarly to Dectin-1.
- 9) The authors apparently assume that *C. albicans* is a representative Dectin agonist. While this organism contains carbohydrates that activate multiple different Dectins, it also activates other macrophage receptors such as the mannose receptor and TLR2/4. Based on this consideration, the statement on p. 6 that, "These data indicate that EPHB2 plays an essential role in Dectin signaling pathway activation in human macrophages, and may function upstream of TAGAP" may not be completely true. It would strengthen the manuscript if the authors would repeat these experiments using specific Dectin-1, 2, and 3 agonists, such as was done in Figs. 1 and 2. Furthermore, the authors should consider the possibility that *C. albicans* directly interacts with EphB2 independently of Dectin-1, similarly to EphA2 (PMID: 29133884)
- 10) A key set of experiments that needs to be performed to support the authors' model that TAGAP functionally interacts with EphB2 and Dectins is to knockdown or knockout TAGAP and then determine the effects of this on the phosphorylation of EphB2 and syk. Similarly, the authors should knockdown or knockout EphB2 and determine the effects on phosphorylation of TAGAP.
- 11) To verify the interaction between Syk and EphB2 in intact cells, the authors should determine in inhibition or knockdown of Syk blocks EphB2 phosphorylation.
- 12) Because *C. albicans* is used as the prototypical stimulus in so many of the figures, an obvious question that should be addressed is whether deletion or knockdown of EphB2 and TAGAP affect the

capacity of macrophages to kill *C. albicans*. Also, it would be useful to know whether the TAGAP^{-/-} mice have increased susceptibility to disseminated or oropharyngeal candidiasis, as would be predicted by the *in vitro* data.

13) In Fig. 4D, there still appears to be significant interaction between Dectin-1 and EphB2 in cells transfected with the KD syk, much more than in cells that were not transfected with any type of syk. How do the authors explain this?

14) In the bar graphs in Fig. 5, it would be better show biological replicates (combined data from multiple mice) rather than technical replicates (combined data from a single mouse). Also, in panel E, the percentage of cells staining for IL-17A is not indicated and the bar graph appears to show that there were 2-fold more IL-17A staining cells in the TAGAP^{-/-} mice than in the control mice. It seems highly probable that this difference is statistically significant, even though it is labeled as not significant.

15) In the dot plots of Fig. 6, it needs to be indicated whether each dot represents a different subject or just a different replicate from the same subject.

16) Dasatenib and vandetanib are nonspecific inhibitors. While they inhibit the phosphorylation of EphB2 and other ephrin receptors, they also inhibit the activity of multiple other tyrosine kinases (see PMID: 20131845). Therefore, no conclusions about the roles of EphB2 and TAGAP in EAE can be made when these inhibitors are used. These experiments should be deleted from the manuscript.

17) In Fig. 7L the diagram indicates that EphB2 and TAGAP play key roles in the activity of Dectin-2 and Dectin-3. However, most of the experiments in the manuscript focused on Dectin-1. Therefore, the interactions with Dectin-2/3 remain speculative.

Reviewer #3, expert in antifungal innate immunity (Remarks to the Author):

The manuscript by Chen et al. describes a newly discovered molecular mechanism that might shed light on the proximal signaling events by a sub-class of PRRs during fungal infections. The authors describe that the tyrosine kinase EPHB2 is recruited by Syk and subsequently phosphorylated by Syk. EPH2 in turn recruits the GAP TAGAP, which then becomes phosphorylated by Syk, thereby forming a scaffold for CARD9. The authors claim that these events occur after Dectin-1 ligation, as well as Dectin-2 and Mincle ligation. The involvement of EPH2 and TAGAP is then touched upon in a setting of Th1 and Th17 differentiation and Th17-driven autoimmune disease EAE.

If this entire mechanism would be proven as well as its role in autoimmune susceptibility, which is linked to SNPs within the TAGAP gene, this would be a very informative and exciting manuscript. However, most of the steps in this mechanism have only been shown in incomplete experiments. For example, while the authors start in fig 3a with showing that dectin-1 and TAGAP co-IP, they quickly conclude that this has to be an indirect association. They go on to introduce the roles for Syk and EPHB2 but they never show that blocking the functions of Syk and/or EPHB2 interferes with the (indirect) association between Dectin-1 and TAGAP. Similar, no SYK and CARD9 co-localization data after EPHB2 knock down. These are just two examples, but basically applies to all experiments. This reviewer needs more direct proof that indeed all these proteins work together as stipulated above. This would greatly impact the importance of the experiments with the EPHB2 inhibitors.

Another more general remark is the use of the term 'dectin signaling pathways'. The authors need to specify per experiment whether they are looking at Dectin-1, Dectin-2 or Mincle. When they use more general stimuli like *C. albicans* or zymosan they can't just use the word 'dectin' in general as it

depends greatly on the *C. albicans* strain which of the above receptors are or are not triggered, while zymosan is a known trigger of TLR2 as well. If they wish to claim that the mechanism they aim to describe is used by all fungi PPRs, then they need a more structured approach to show the results with all different and receptor-specific stimuli.

The manuscript lacks a description on why a GAP protein (TAGAP) would serve as a scaffold for CARD9. What about its GAP activity, is there no need for this in the Dectin-1 signaling pathway?

Why are almost all Th differentiation assays focused on autoimmunity and not anti-fungal immunity, which would link better with the use of the fungal-specific ligands in the first four figures? Wouldn't the data (molecular mechanism vs role TAGAP/EBHP2 in Th17/EAE) be better appreciated if separated into two manuscript that can go more in depth on both fronts?

More specific comments:

- explain why expression of certain genes are examined, like IL-2 and CXCL1? What is their role in innate/adaptive immune responses?
- use the correct terms for examined genes: IL-12 should be IL-12a, IL-23 should be IL-23a etc.
- why is IL-1b examined in Fig. 2c while IL-2 is used in Fig. 2a?
- if indeed dectin-1 and TAGAP can associate indirectly with each other in HEK293 cells, show that these cells express EPHB2.
- PLCg2 is not a kinase
- Fig. 1d uses the same *Candida* as Fig. 1f? labeled differently in figure.
- Fig. 2b levels of HSP90 are significantly lower in TAGAP^{-/-} cells
- Shouldn't wild-type mice be named TAGAP^{+/+} instead of TAGAP^{+/-}?
- What cell line is used in Fig. 3c?
- What happened with the expression of the HA-TAGAP after the HA IP/HA IB in the last lane of Fig. 4e?
- Fig 4k shows co-localization of SYK and CARD9 in TAGAP^{-/-}
- Fig 5a shows only a marginal effect of TAGAP deficiency on Th1 differentiation; how do the authors explain this since there was a significant effect on IL12a expression.
- An explanation is needed for the DCs results – TAGAP deficiency compromises the ability of DCs to polarize Th17 differentiation, however DCs only show low TAGAP expression in Fig. 1a compared to macrophages. In the same line, what about EPHB2 expression in DCs? In macrophages it needs to be induced before stimulation with *C. albicans* has a stimulatory effect – does this not need to happen in DCs?
- the authors need to explain how the SNPs influence TAGAP mRNA expression. Does the difference in AA change the expression (first SNP)? Is the second SNP located in the promoter region of TAGAP?
- please confirm the differences in TAGAP mRNA levels between the different SNPs on the protein level.

Point-by-point response

We are really thankful for all of three reviewer's comments and suggestions, and many of them are really helpful and constructive, which helped us to further validate the role of EPHB2-TAGAP in the anti-fungi signaling pathway and T cells polarization. Please find the point-by-point response below:

Reviewers' comments:

Reviewer #1, expert in Th17 (Remarks to the Author):

Chen et al. demonstrated that TAGAP and EPHB2 were critical in dectin ligands-induced production of proinflammatory cytokines and its signal transduction in macrophages. EPHB2 associates and phosphorylates TAGAP, then phospho-TAGAP can bind to CARD9 to activate MAPK and NFkB pathways. TAGAP deficient mice showed ameliorated EAE symptom, and the authors concluded that reduced Th17 responses in TAGAP deficient mice was due to the incomplete production of dectin-dependent cytokines from macrophages. Although discovery of the function of TAGAP and EPHB2 in dectin-signaling is provocative, several critical issues are of concern. The authors effort to eliminate these concerns is of great importance.

Major concerns

1. Impaired EAE symptom in TAGAP-deficient mice has previously reported (Tamehiro et al, Immunol Cell Biol., 95: 729, 2017), showing that T cell intrinsic defect of Th17 differentiation. Indeed, effect of TAGAP deficiency in Th17 differentiation from naive T cells in vitro (Fig. 5e) was completely opposite from the reported results. The authors should quote the paper and reconcile or discuss about the discrepancy. Because obvious contrary indication exists, the author should show more persuasive results in order to insist that reduction of Th17 induction in vivo was due to the dectin/EPHB2/TAGAP-dependent, impaired cytokine production from macrophages.

We thank reviewer's suggestion, and we quoted this paper and discussed about the possible reason for the discrepancy in the Discussion part of revised version, which is as following: "Interestingly, one study found that TAGAP deficient CD4+ T cells had an intrinsic defect in Th17 polarization⁴². We didn't find the in vitro Th17 differentiation defect in TAGAP-deficient CD4+ T cells, and this was further confirmed by EAE model on Rag2-deficient mice transferred with CD4+ T cells from control or TAGAP-deficient mice (Figure 5E and Figure S7). One of possible reason for this discrepancy could be due to the different T cell polarization methods used between us, since we noticed that the other group polarized Th17 cells in vitro by adding TGFβ, IL-6, IL-1β and IL-23a in the presence of anti-CD3, anti-CD28, anti-IFN-γ and anti-IL-4, while we didn't add IL-1β and IL-23a for Th17 cell in vitro polarization, as many studies have found that IL-6 plus TGFβ were sufficient for naive CD4+ cell polarization to Th17 cells in vitro^{43,44,45}. However, studies found that IL-23a didn't further expand Th17 cells in an in vitro

polarization condition, instead induce GM-CSF production in existing Th17 cells, and that GM-CSF had an essential role in their encephalitogenicity^{46,47}. Although there may be different ways of polarization Th17 cells in vitro, our EAE model on Rag2-deficient mice clearly indicates that TAGAP-deficient CD4+ T cells didn't have any intrinsic defect (Figure S7)."

for example:

Transfer of wild type macrophages into TAGAP-deficient mice can restore reduced EAE?

Dectin (or EPHB2) deficient mice show impaired EAE or Th17 induction?

We addressed this concern by setting up EAE model by using Rag2 KO mice transferred with CD4+ T cells from control or TAGAP-deficient mice (this experiment was repeated independently by professor Cunjin-Zhang at Nanjing University). We didn't find significant difference in terms of EAE severity and the CD4+ T cell differentiation *in vivo*, which further proved that TAGAP didn't play an intrinsic role in T cell differentiation (Figure S7).

Minor points

2. In Fig.5, there are several mistakes? in the number of % in quadrant. The % written are in some cases different from the one written in raw data. (5a left top 0.74%>1.44%, also in 5e and 5f). This kind of mistake tends to give negative impression about reliability of the author's data management.

We thank reviewers' comments, and the top-right panel in 5a represented data from spleen, and lower-right panel in 5a represented the data from lymph nodes. We re-plotted the data by individual dots to make the data more transparent (each dot represents the data from an individual mouse).

In the revised version, we combined the data from two experiments to make the difference more significance, which includes 8 pairs of control and TAGAP-KO mice for Figure 5a, and 10 pairs of mice for Figure 5e and 6 pairs of mice for 5f.

Comments

3. Since involvement of TAGAP and EPHB2 in dectin-signaling was firstly shown in the manuscript, one would be more interested in knowing if TAGAP-deficient mice would be sensitive for fungal/yeast infection such as *Pneumocystis carinii* or *Candida albicans*, which was seen in dectin1/2-deficient mice.

We are thankful for the reviewer's suggestion, and we did the fungi infection model by using control and TAGAP-deficient mice. We found that TAGAP-deficient mice were much more susceptible to *C. albicans* infection than control mice, and we included the data in the Figure S5 in the revised version.

Reviewer #2, expert in antifungal innate immunity (Remarks to the Author):

This manuscript describes the role of TAGAP and EphB2 in macrophage activation, mainly in response to heat-killed *C. albicans*. It contains a very large amount of data, much of which are convincing and support some of the conclusions. Specific comments:

1) While the manuscript is easy to comprehend, its English could be improved by a thorough editing to correct the numerous minor grammatical errors. It would also make the manuscript significantly easier to review if the lines were numbered and if the figure legends were located near the figure

We are thankful for the reviewer's comments, and we improved the English writing of this manuscript by ask a friend (Bradley Martine at Harvard Medical College) to correct the errors and polish the language.

We are thankful for the reviewer's suggestions, and we numbered the lines and put the Figures and Figure Legends together in the revised manuscript.

2) In the Western blots that show the phosphorylation of specific proteins, such as in Figures 1b, d, f , 2 b and d, the total amount of unphosphorylated proteins should also be shown. For example, in addition to probing the blots for p-ERK, the amount of total ERK should also be shown. Also, in Fig 1d, the macrophages in the zymosan experiments appear to be constitutively activated because ERK is highly phosphorylated even at time 0. It would also strengthen the results if densitometry data from replicate Western blots were shown in the supplementary data.

We are thankful for the reviewer's comments, and we included the bands of unphosphorylated proteins as additional control. We also did the densitometry analysis for phosphorylated proteins, and included it in Figure. S9.

[Redacted]

3) In Fig. 1 e and 2a and c, the real-time PCR data from all 3 experiments should be combined rather than showing the results of single experiment.

We thank the reviewer's suggestion, and original data of Figure 1C, 1E, 2A and 2D were from 2 experiments, and now we included data from 3 combined experiments in Figure 1C, 1E, 2A and 2D in the revised version.

4) Why was heat-killed *C. albicans* used the stimulation experiments rather than live organisms?

We thank reviewer's comments. Heat killing of *C. albicans* resulted in exposure of β -glucans on the surface of the cell wall and subsequent recognition by dectin-1, whereas live yeasts stimulated monocytes/macrophages mainly via recognition of cell-surface mannans, which will activate dectin-2/3 pathway (Gow NA, *et al.* J Infect Dis. 2007 Nov 15;196(10):1565-71). We used heat-killed *C. albicans* to activate Dectin-1 signaling pathway.

5) RNA-seq data should be deposited in a public database.

We already deposited the raw RNA-seq data into public database "figshare", and we provided the username and password in the manuscript for the accessing of these data.

6) The Western blotting data showing the differences in phosphorylation of IKK α and ERK in Fig. 2 are not very convincing. Again, total IKK α and ERK should be shown and densitometry results of replicate Western blot should be presented.

We thank the reviewer's suggestions, and we replaced the western blot data of Figure 2 with better representative result from an independent experiment. We included total proteins as control, and also did the densitometry from replicated western blot (Figure S9).

7) The description of the mass spectrometry experiments whose results are shown in Fig. 3b is so brief that the reader cannot understand what was actually done. What cells were used and what protein was the target of the immunoprecipitation? Was it TAGAP? If so, were any Dectins detected?

We thank the reviewer's comments, and we are sorry not describing the experiment very clearly. We detailed describe the mass spectrometry experiment in figure legend and Methods. We did the experiment in Flag-TAGAP stable-transfected Thp1 cells, and immunoprecipitated TAGAP by using anti-Flag antibody. We then eluted the proteins and sent the eluted protein to company to identify the TAGAP-interacting proteins by mass spectrometry.

We didn't detect Dectins in the mass spectrometry identification, which may be due to the low-expression level of Dectin proteins in the Thp1 cells (Liangkuan Bi, *et al.* THE JOURNAL OF BIOLOGICAL CHEMISTRY VOL. 285, NO. 34, pp. 25969–25977, August 20, 2010).

8) It would strength the IP data if it could be shown that TAGAP interacts with Dectin-1 and EphB2 in either mouse or human macrophages rather than in transfected HEK cells.

We thank the reviewer's comments, and we performed endogenous immunoprecipitation by using CARD9 antibody in heat killed *C. albicans*-stimulated Thp1 cells. We found that the endogenous interaction between CARD9, TAGAP and EPHB2, and the interaction became stronger after heat-killed *C. albicans* stimulation (Figure 4K).

Also, the authors should determine whether knockdown of EphB2 blocks the association of Dectin-1 and TAGAP, as suggested by their model. If antibodies are not available for

performing immunoprecipitation of untagged proteins, then the authors could at least perform proximity ligation assays in intact cells.

We thank the reviewer's suggestion, and we made EPHB2 knocked down 293T cells by CRISPR-Cas9 (Figure S3E-F). We found that the interaction between TAGAP and Dectin-1/2 was significantly decreased, which suggests that the interaction between TAGAP and Dectins was mediated by EPHB2, at least partially. The existence of residual interaction between TAGAP and Dectin-1/2 in EPHB2 knocked down cells may be mediated by the residual EPHB2. Another possibility is that besides EPHB2, there may be other molecules which also mediated the interaction between TAGAP and Dectins.

In addition, it would be useful to know if Dectin-2 and 3 acted similarly to Dectin-1.

We did the IP between Dectin-2/3 and TAGAP/EPHB2, and found that TAGAP and EPHB2 can bind to Dectin-2, but not Dectin-3 (Figure S3A, S3D and S3F), which suggests that TAGAP and EPHB2 function in Dectin-2/3 signaling pathway through Dectin-2.

9) The authors apparently assume that *C. albicans* is a representative Dectin agonist. While this organism contains carbohydrates that activate multiple different Dectins, it also activates other macrophage receptors such as the mannose receptor and TLR2/4. Based on this consideration, the statement on p. 6 that, "These data indicate that EPHB2 plays an essential role in Dectin signaling pathway activation in human macrophages, and may function upstream of TAGAP" may not be completely true.

We thank the reviewer's suggestions, and we changed the statement in the manuscript as the following: "These data indicate that EPHB2 plays an essential role in anti-fungi signaling pathway in human macrophages".

It would strengthen the manuscript if the authors would repeat these experiments using specific Dectin-1, 2, and 3 agonists, such as was done in Figs. 1 and 2.

We thank the reviewer's suggestions, and we examined the role of EPHB2 in Dectin-1, 2, 3 ligands-induced signaling pathways (Figure 3G-I), and found that EPHB2 plays a critical role in all of the Dectin ligands-induced signaling pathway activation.

Furthermore, the authors should consider the possibility that *C. albicans* directly interacts with EPHB2 independently of Dectin-1, similarly to EphA2 (PMID: 29133884)

We thank the reviewer's suggestion, and we were actually quite curious to explore whether EPHB2 can directly bind β -glucan or α -Mannan as EphA2. In the current study, we found that EPHB2 can bind Dectin-1 and Dectin-2, and the interaction between Dectin-1 and EPHB2 can be increased after co-expression of Syk, and EPHB2 can be phosphorylated by Syk (Figure S3C, S3D, 4C and 4D). These data suggest that EPHB2 function downstream of Syk in the Dectin pathways, although we can't exclude the possibility that EPHB2 directly binds to yeast and activates anti-fungi signaling pathway as a parallel pathway. While we think this may be beyond the scope of this study, since in this study, we want to mainly focus on the functional role of TAGAP in the anti-fungi signaling pathway, and subsequently the role of TAGAP in the T helper cells polarization, which leads to autoimmune diseases susceptibility.

Actually, we have another project ongoing which is to detailly explore the functional role of EPHB2 in the anti-fungi signaling pathway, and we are examining whether EPHB2 can directly bind to β -glucan or α -Mannan, while we want to present this part of data in the future as an independent study.

10) A key set of experiments that needs to be performed to support the authors' model that TAGAP functionally interacts with EphB2 and Dectins is to knockdown or knockout TAGAP and then determine the effects of this on the phosphorylation of EphB2 and syk.

We thank the reviewer's suggestions, and we examined the phosphorylation of upstream kinases such as syk and Raf-1 in control or TAGAP-deficient BMDMs, and we included this data in the revised version (Figure 1G). We didn't find phosphorylation of syk and Raf-1 has any defect in TAGAP-deficient cells after heat killed *C. albicans* stimulation, which indicates that TAGAP functions downstream of SYK and Raf-1. We didn't find significant defect of p-EPHB2 in TAGAP-knocked down Thp1 cells compared to that in control cells after heat-killed *C. albicans* stimulation, which indicates that TAGAP functions downstream of EPHB2 in Dectin-1 signaling pathway (Figure S3H).

Similarly, the authors should knockdown or knockout EphB2 and determine the effects on phosphorylation of TAGAP.

We thank the reviewer's suggestions, and we did the experiment, and included the data in the revised manuscript (Figure S3I). We found that the phosphorylation of TAGAP was abolished in EPHB2-knocked down Thp1 cells, which indicates that EPHB2 is required for the phosphorylation of endogenous TAGAP. Interestingly, we found that TAGAP was phosphorylated even at basal level, which suggests that the TAGAP and EPHB2 constantly bind together, and this data is consistent with our endogenous IP data (Figure 4K).

11) To verify the interaction between Syk and EphB2 in intact cells, the authors should determine in inhibition or knockdown of Syk blocks EphB2 phosphorylation.

We thank the reviewer's suggestions, and we used syk inhibitor Piceatannol to block syk kinase activity, and examined the p-EPHB2 in Thp1 cells (Figure S3G). The phosphorylation of EPHB2 was almost abolished in Thp1 cells after Piceatannol treatment, which indicates that the endogenous phosphorylation of EPHB2 is entirely dependent on syk.

12) Because *C. albicans* is used as the prototypical stimulus in so many of the figures, an obvious question that should be addressed is whether deletion or knockdown of EphB2 and TAGAP affect the capacity of macrophages to kill *C. albicans*.

We are thankful for the reviewer's suggestion, and in this study, we didn't use live *C. albicans* for the stimulation of the cells (we used heat killed *C. albicans* to stimulate Dectin-1 signaling). In this study, we are trying to explore the functional role of EPHB2-TAGAP axis in the innate anti-fungi signaling pathway, which produce cytokines such as IL-23a and IL-12a for T cell differentiation, and we tried to understand how the dysregulation of this innate anti-fungi pathway lead to autoimmune disease susceptibility. We think that killing of yeasts by macrophages is sort of an independent pathway in innate immune anti-fungi immunity, and we are actually exploring the role of TAGAP and EPHB2 in yeast killing by macrophages as an independent study, and we prefer to show this part of the data in this future as an independent study.

Also, it would be useful to know whether the TAGAP^{-/-} mice have increased susceptibility to disseminated or oropharyngeal candidiasis, as would be predicted by the in vitro data.

We thank reviewer's suggestion. We did the fungal infection model by using TAGAP-deficient

mice, and found that TAGAP-deficient mice were more susceptible to *C. albicans* infection compared to control mice (Figure S5).

13) In Fig. 4D, there still appears to be significant interaction between Dectin-1 and EphB2 in cells transfected with the KD syk, much more than in cells that were not transfected with any type of syk. How do the authors explain this?

We are thankful for the reviewer's comments. Transfection of wild-type syk increased the interaction between Dectin-1 and EPHB2 compared to the transfection of empty vector, and a possible explanation for this data is that phosphorylated EPHB2 gets higher affinity with syk, which forms a complex with Dectin-1 receptor. Compared to the transfection of empty vector, the recruitment of EPHB2 to Dectin-1 complex was also increased after transfection of syk KD, and this data may suggest that un-phosphorylated EPHB2 can still bind syk, although the binding affinity may be weaker compared to the interaction between p-EPHB2 and syk.

14) In the bar graphs in Fig. 5, it would be better show biological replicates (combined data from multiple mice) rather than technical replicates (combined data from a single mouse).

We are thankful for the reviewer's suggestions. The original data is combined data from multiple mice, for example, 4 pairs of mice for Figure 5A (as one representative result).

Also, in panel E, the percentage of cells staining for IL-17A is not indicated and the bar graph appears to show that there were 2-fold more IL-17A staining cells in the TAGAP^{-/-} mice than in the control mice. It seems highly probable that this difference is statistically significant, even though it is labeled as not significant.

We are thankful for the reviewer's comments. The original Figure 5E data is from 4 pairs of control and TAGAP KO mice, and the Th17 cells percentages between control and KO mice didn't reach significance (please see below). We independently repeated this experiment three times (we used 3 pairs of mice in the last two experiments), and please find the data below. We didn't find that TAGAP KO naïve CD4⁺ T cells have a differentiation defect. We combined the data from all three times in the revised version.

15) In the dot plots of Fig. 6, it needs to be indicated whether each dot represents a different subject or just a different replicate from the same subject.

We are thankful for the reviewer's comments. Each dot in Figure 6 represents the data got from each individual, not the replicate from the same individual, and we included more detailed description in the Figure legends.

16) Dasatenib and vandetanib are nonspecific inhibitors. While they inhibit the phosphorylation of EphB2 and other ephrin receptors, they also inhibit the activity of multiple other tyrosine kinases (see PMID: 20131845). Therefore, no conclusions about the roles of EphB2 and TAGAP in EAE can be made when these inhibitors are used. These experiments should be deleted from the manuscript.

We are thankful for the reviewer's comments. We agree with the reviewer that Dasatenib and vandetanib are not specific to only target EPHB2. But we still think that this data has its value and should be kept in the manuscript, and the reasons are 1) Almost all of the small compound inhibitors have off-target effects by targeting other proteins, and this is almost inevitable; 2) Our data clearly indicates that both of the inhibitors can block kinase activity of EPHB2, and can inhibit Dectin-1/2/3 ligands induced signaling and gene expression (Figure 7E-L and Figure S7B); 3) More importantly, our EAE data clearly indicates that both Dasatenib and vandetanib (especially vandetanib) can significantly attenuate mice EAE severity by decreasing the T helper cells polarization and brain infiltration (Figure 7J-K), which suggests that these two proved drugs could have potential for treating autoimmune diseases such as multiple sclerosis. From the disease treatment perspective, although these two drugs may not have good specificity, as long as they have the potential for the treatment of human disease, we believe that keeping this data in the manuscript is appropriate; 4) There are many examples that "old" drugs work for new indications due to identification of new target proteins of the drugs (1. Dai J, *et al.* Cell. 2019 Mar 7;176(6):1447-1460; 2. Skrott Z, *et al.* Nature. 2017 Dec 14;552(7684):194-199; 3. Huber KV, *et al.* Nature. 2014 Apr 10;508(7495):222-7).

17) In Fig. 7L the diagram indicates that EphB2 and TAGAP play key roles in the activity of Dectin-2 and Dectin-3. However, most of the experiments in the manuscript focused on Dectin-1. Therefore, the interactions with Dectin-2/3 remain speculative.

We are thankful for the reviewer's comments. We provided evidence that both TAGAP and EPHB2 play a critical role in the Dectin-2/3 ligands-induced signaling pathway in the revised manuscript (Figure 2 and Figure 3G-J).

We also examined the interaction between TAGAP/EPHB2 and Dectin-2/3 (Figure S3A, S3D and S3E). Interestingly, we found that TAGAP can bind Dectin2 but not Dectin-3, which suggests that the functional role of TAGAP and EPHB2 in Dectin-2/3 signaling pathway is through Dectin2.

Reviewer #3, expert in antifungal innate immunity (Remarks to the Author):

The manuscript by Chen et al. describes a newly discovered molecular mechanism that might shed light on the proximal signaling events by a sub-class of PPRs during fungal infections. The authors describe that the tyrosine kinase EPHB2 is recruited by Syk and subsequently

phosphorylated by Syk. EPH2 in turn recruits the GAP TAGAP, which then becomes phosphorylated by Syk, thereby forming a scaffold for CARD9. The authors claim that these events occur after Dectin-1 ligation, as well as Dectin-2 and Mincle ligation. The involvement of EPH2 and TAGAP is then touched upon in a setting of Th1 and Th17 differentiation and Th17-driven autoimmune disease EAE.

If this entire mechanism would be proven as well as its role in autoimmune susceptibility, which is linked to SNPs within the TAGAP gene, this would be a very informative and exciting manuscript. However, most of the steps in this mechanism have only been shown in incomplete experiments. For example, while the authors start in fig 3a with showing that dectin-1 and TAGAP co-IP, they quickly conclude that this has to be an indirect association. They go on to introduce the roles for Syk and EPHB2 but they never show that blocking the functions of Syk and/or EPHB2 interferes with the (indirect) association between Dectin-1 and TAGAP. Similar, no SYK and CARD9 co-localization data after EPHB2 knock down. These are just two examples, but basically applies to all experiments. This reviewer needs more direct proof that indeed all these proteins work together as stipulated above. This would greatly impact the importance of the experiments with the EPHB2 inhibitors.

We are thankful for the reviewer's comments, and we realize that the evidence for some of the mechanism which we proposed in the manuscript wasn't quite strong. We did the experiments which suggested by the reviewer, which was show in the Figure S3E-F and Figure S8. We found that the interaction between TAGAP and Dectin-1/2 was significantly decreased in EPHB2-knocked down cells, which suggests that the interaction between TAGAP and Dectins was mediated by EPHB2, at least partially. The reason of existence of residual interaction between TAGAP and Dectin-1/2 in EPHB2 knocked down cells may be mediated by the residual EPHB2. Another possibility is that besides EPHB2, there may be other molecules which also mediated the interaction between TAGAP and Dectins (Figure S3E-F). After EPHB2 inhibitor treatment, the co-localization of syk and card9 was abolished in Thp1 cells, which further prove the critical role of EPHB2 kinase activity in Dectin ligand-induced signaling pathway (Figure S8).

[Redacted]

Another more general remark is the use of the term 'dectin signaling pathways'. The authors need to specify per experiment whether they are looking at Dectin-1, Dectin-2 or Mincle. When they use more general stimuli like *C. albicans* or zymosan they can't just use the word 'dectin' in general as it depends greatly on the *C. albicans* strain which of the above receptors are or are not triggered, while zymosan is a known trigger of TLR2 as well. If they wish to claim that the mechanism they aim to describe is used by all fungi PRRs, then they need a more structured approach to show the results with all different and receptor-specific stimuli. We are thankful for the reviewer's comments, and we thoroughly examined the manuscript, and changed 'dectin signaling pathways' to more general term as "anti-fungi signaling

pathway”, and some of them changed to more specific term as “Dectin-1, Dectin-2 or Mincle ligands-induced signaling pathway”.

We used heat-killed *C. albicans* to stimulate mouse and human macrophages, as heat killed *C. albicans* resulted in exposure of β -glucans on the surface of the cell wall and subsequent recognition by dectin-1 (Gow NA, et al. J Infect Dis. 2007 Nov 15;196(10):1565-71). We didn't use live *C. albicans* to stimulate the cells.

We used D-zymosan to stimulate macrophages in this study, as D-zymosan was believed to be a Dectin-1 ligand (Figure 1D). We didn't use zymosan to stimulate cells.

The manuscript lacks a description on why a GAP protein (TAGAP) would serve as a scaffold for CARD9. What about its GAP activity, is there no need for this in the Dectin-1 signaling pathway?

We are thankful for the reviewer's comments. In this study, our data suggests that TAGAP functions as an adaptor, which mediated the signaling transduction through recruiting downstream CARD9 to upstream molecule, such as syk and EPHB2 (Figure 4E, 4F, 4I-M). We found that the GAP domain of TAGAP is required for its binding to EPHB2, which indicates that binding of EPHB2 by TAGAP is through GAP domain (Figure 3D). We are now exploring whether small G proteins, such as Rac1, RhoA and cdc42 are also involved in TAGAP-mediated anti-fungi signaling pathway, since GAP domain is supposed to regulate activity of small G proteins. Since TAGAP-small G protein axis may represents an independent signaling pathway, we prefer to explore it as an independent study.

Why are almost all Th differentiation assays focused on autoimmunity and not anti-fungal immunity, which would link better with the use of the fungal-specific ligands in the first four figures? Wouldn't the data (molecular mechanism vs role TAGAP/EBHP2 in Th17/EAE) be better appreciated if separated into two manuscript that can go more in depth on both fronts? We are thankful for the reviewer's comments, and we really appreciated the review's suggestion. The initial aim of this study was to try to understand the molecular and cellular mechanism of TAGAP polymorphisms to the susceptibility of many autoimmune diseases, such as MS. In other words, initially, we wanted to explore the signaling pathway in which TAGAP is involved, and how the dysregulation of this signaling pathway leads to autoimmune diseases susceptibility.

Actually, we were surprised at beginning when we first found that TAGAP play an essential role in the innate anti-fungus signaling pathway, while now we think that it's quite interesting, since it may provide an example to demonstrate how the dysregulation of innate immune pathway lead to autoimmune diseases susceptibility.

[Redacted]

More specific comments:

-explain why expression of certain genes are examined, like IL-2 and CXCL1? What is their role in innate/adaptive immune responses?

We are thankful for the reviewer's comments. We mainly examined two types of genes

induced by Dectin ligands stimulation: one group is IL-2, IL-12a and IL-23a, which were known to play a critical role for T helper cells proliferation and differentiation (Th1 and Th17); another group of genes which we examined is proinflammatory gene, such as IL-1 β , IL-6 and CXCL1, and this group of cytokine gene was known to cause inflammation and recruitment of downstream inflammatory cells, such as neutrophils. Although IL-1 β and IL-6 were also known to play a role to polarize Th17 cells differentiation.

-use the correct terms for examined genes: IL-12 should be IL-12a, IL-23 should be IL-23a etc.

We are thankful for the reviewer's comments, and we have examined the manuscript thoroughly and corrected it.

-why is IL-1b examined in Fig. 2c while IL-2 is used in Fig. 2a?

We are sorry for the inconsistency. We presented that data of IL-2 in Figure 2c in the revised version.

-if indeed dectin-1 and TAGAP can associate indirectly with eachother in HEK293 cells, show that these cells express EPHB2.

We are thankful for the reviewer's comments, and please find the data in Figure S3E-F.

-PLCg2 is not a kinase

We are sorry for the mistake, and we corrected it in the revised manuscript.

-Fig. 1d uses the same Candida as Fig. 1f? labeled differently in figure.

We are thankful for the reviewer's comments. The *C. albicans* used in Figure 1D and 1F is the same strain, and both are heat-killed SC-5314 strain of *C. albicans*. We are sorry not clarify it clearly in the manuscript, and we have kept the labeling consistent in the revised version.

-Fig. 2b levels of HSP90 are significantly lower in TAGAP-/- cells

We are thankful for the reviewer's comments, and we have chosen another better representative data for Figure 2B from an independent experiment.

-Shouldn't wild-type mice be named TAGAP +/+ instead of TAGAP +/-?

We used TAGAP heterozygous littermate mice as the control for TAGAP-KO mice. Our early study found that the cells from TAGAP+/+ and the cells from TAGAP+/- showed similar response to heat-killed *C. albicans* stimulation.

-What cell line is used in Fig. 3c?

We are sorry not labeling it clearly. Original Figure 3C is done in human mono cell line U937, and to keep consistent (most of the data from Figure 3 were done by using Thp1), we now move the original Figure 3C to Figure S3B, and present the data from Thp1 in Figure 3C in the revised version.

-What happened with the expression of the HA-TAGAP after the HA IP/HA IB in the last lane

of Fig. 4e?

We are thankful for the reviewer's comments. In this experiment, the immunoprecipitated HA-TAGAP was less in the last group compared to the group which co-expression with wild-type EPHB2 and TAGAP. We repeated this experiment several times, and our data demonstrated that wild-type EPHB2 can phosphorylate TAGAP, while the kinase-dead EPHB2 can't, and please find the data from another independent experiment below.

-Fig 4k shows co-localization of SYK and CARD9 in TAGAP -/-

We are thankful for the reviewer's comments. We repeated this experiment several times, and there is always a strong co-localization between syk and card9 in control macrophages after heat-killed *C. albicans* stimulation, while the co-localization of syk and card9 was almost abolished in TAGAP-deficient cells. There may be marginal co-localization between syk and card9 in TAGAP-deficient cells, while we think compared to the co-localization in the control cells, the co-localization was almost abolished.

We changed the description for this part in the revised manuscript as "while the co-localization between SYK and CARD9 was almost abolished in TAGAP-deficient macrophages after heat-kill *C. albicans* stimulation".

-Fig 5a shows only a marginal effect of TAGAP deficiency on Th1 differentiation; how do the authors explain this since there was a significant effect on IL12a expression.

We are thankful for the reviewer's comments. In the Figure 5A, we used naïve mice to do the experiment (these mice didn't get immunization), and the mice do show comparable Th1 cells population in the spleens between control and TAGAP-deficient mice (there is higher Th1 cells percentages in the lymph nodes of TAGAP-deficient mice compared to control mice, although the difference didn't reach significance). The mice were raised under SPF condition, and there are some microorganisms such as bacterial or fungus species exist in the SPF condition which can drive CD4+ T cells to Th1 and Th17 cells. The Th17 and Th1 cells which we found in these naïve mice should be polarized by the microorganisms in the SPF condition (1. Sano T, *et al.* Cell. 2015 Oct 8;163(2):381-93. doi: 10.1016/j.cell.2015.08.061; 2. Sun M, *et al.* Nat Commun. 2018 Sep 3;9(1):3555. doi: 10.1038/s41467-018-05901-2). A possible explanation for the phenotype of comparable Th1 cell between control and TAGAP-deficient mice may be because that the Th1 polarization microorganism in our SPF condition isn't through Dectin signaling pathways, so there will be no defect of Th1 cells differentiation in TAGAP-KO mice.

We are pretty sure that the Th1 cell polarization also has a defect in TAGAP-KO mice after MOG immunization with the use of adjuvant such as heat-killed Mycobacterium tuberculosis

H37Ra or heat-killed yeast *C. albicans*, which is known to activate Dectin-1 signaling pathway (Figure 5B-C).

-An explanation is needed for the DCs results – TAGAP deficiency compromises the ability of DCs to polarize Th17 differentiation, however DCs only show low TAGAP expression in Fig. 1a compared to macrophages. In the same line, what about EPHB2 expression in DCs? In macrophages it needs to be induced before stimulation with *C. albicans* has a stimulatory effect – does this not need to happen in DCs?

We are thankful for the reviewer's comments. We examined the Dectin-1 and Dectin-2/3 ligands-induced signaling activation in BMDCs from control or TAGAP-deficient mice (Figure S2A-S2D). Interestingly, TAGAP-deficient BMDCs also shows defect signaling activation and proinflammatory gene expression, which suggests that EPHB2-TAGAP axis may also exist in DCs, and we will further explore it in the future.

Since the human monocytes-DC cell line is not available, we can't perform the same experiment as did in Thp1 and U937 cells (Figure 3C and Figure S3B). However, we did examine the EPHB2 expression during mouse BMDCs differentiation after GM-CSF and IL-4 priming, and we didn't find induction of EPHB2 protein during the BMDCs differentiation. Interestingly, it seems that there is modification of EPHB2 after GM-CSF and IL-4 priming (indicated by the asterisk). Since this study is mainly focused on macrophages, and we will further investigate the role of EPHB2 in dendritic cells in the future (please find the data below).

- the authors need to explain how the SNPs influence TAGAP mRNA expression. Does the difference in AA change the expression (first SNP)? Is the second SNP located in the promoter region of TAGAP?

We are sorry not clarifying it clearly in the manuscript. The first SNP rs1738074 is localized in the first exon of TAGAP gene, and the transcription of TAGAP starts from second exon, so it won't cause protein-coding change of TAGAP. The second SNP rs3127214 is localized in the promoter region of the TAGAP gene. We found that these two snps affect TAGAP mRNA expression, and we believe that these two snps mainly affect the transcription of TAGAP mRNA, which may affect the binding of specific transcriptional factor to the promoter.

- please confirm the differences in TAGAP mRNA levels between the different SNPs on the protein level.

We are thankful for the reviewer's suggestions. Although we do really want to validate the TAGAP protein level by using the PBMC samples from individuals who carries TAGAP different genotypes (Figure 6A), while conducting this experiment is very difficult: we can only get about 1ml whole blood sample of healthy volunteers from our collaborators in the hospital, and we isolated PBMCs and separate them into three groups to do the experiments for the Figure 6. Each group may only contain about 1 million cells, which is almost impossible to do

the western blot for TAGAP detection due to the sensitivity of the TAGAP antibody. We actually tried to detect TAGAP protein in human PBMCs from 1 ml whole blood sample, but we couldn't detect any signal for TAGAP.

Overall, we are grateful for all of three reviewer's efforts on this manuscript, which really helped us to further validate the functional role of EPHB2-TAGAP in the anti-fungi immune response and the role in T cells polarizations.

REVIEWERS' COMMENTS:

Reviewer #1 (Remarks to the Author):

My concern about the manuscript has been mostly addressed by the authors.

Reviewer #2 (Remarks to the Author):

This manuscript is much improved, but still has issues that should be addressed, especially the concern that the data with dasatanib and vandetanib were not interpreted correctly. Specific comments:

1. As mentioned in the previous critiques, dasatinib and vandetanib are broad-spectrum tyrosine kinase inhibitors. Therefore, it is not correct to call them EPHB2 inhibitors, as is stated in lines 7 and 8 in the abstract. I suggest that the word "EPHB2" be replaced with the words "broad-spectrum tyrosine kinase."
2. Also, on P. 11, lines 13-15, it needs to be stated that the drugs inhibit EPHB2 and multiple other kinases.
3. Because of the broad-spectrum nature of these kinase inhibitors, the statement in lines 22-23 of P. 11 needs to be qualified. While the data indicate that inhibition of some kinase blocks syk and CARD9 co-localization, it cannot be definitively concluded that this is due to inhibition of EPHB2. Similarly, the first sentence in the legend of Fig. 7 is not accurate because it cannot be determined whether the effects of the drugs are due to inhibition of EPHB2 or some other kinase.
4. Again, the statement on lines 12-13 on P. 13 is not accurate and needs to be qualified to state that the dasatinib and vandetanib do not definitely link inhibition of EPHB2 to the outcome of EAE because they inhibit so many kinases.
5. Throughout the manuscript, the authors use the words, "anti-fungi signaling." The correct terminology is "anti-fungal signaling."
6. The Western blots in Figs. 1G and S2A are not of publication quality. In Fig. 1G it is difficult for this reviewer to see if there is any significant phosphorylation of Raf or syk in any of the cells at any time point. In Fig. S2A, the phosphorylation of ERK appears to be constitutive.
7. P. 8 line 13, "medicated" should be changed to "mediated."

Point-by-point response

We are really thankful for the reviewer's comments and suggestions. Please find the point-by-point response below:

Reviewers' comments:

Reviewer #1 (Remarks to the Author):

My concern about the manuscript has been mostly addressed by the authors.

We are thankful for the reviewer's comment.

Reviewer #2 (Remarks to the Author):

This manuscript is much improved, but still has issues that should be addressed, especially the concern that the data with dasatanib and vendetanib were not interpreted correctly. Specific comments:

We are thankful for the reviewer's comment, and we changed the "EPHB2 inhibitor" to "broad-spectrum tyrosine kinase inhibitors" throughout the manuscript.

1. As mentioned in the previous critiques, dasatanib and vendetanib are broad-spectrum tyrosine kinase inhibitors. Therefore, it is not correct to call them EPHB2 inhibitors, as is stated in lines 7 and 8 in the abstract. I suggest that the word "EPHB2" be replaced with the words "broad-spectrum tyrosine kinase."

We are thankful for the reviewer's comment, and we depleted the "EPHB2 inhibitors" in the abstract of the revised manuscript.

2. Also, on P. 11, lines 13-15, it needs to be stated that the drugs inhibit EPHB2 and multiple other kinases.

We are thankful for the reviewer's comment, and we changed the description to "broad-spectrum tyrosine kinase inhibitors Dasatinib and Vandetanib" in the lines 30-31 of P. 10 in the revised manuscript.

3. Because of the broad-spectrum nature of these kinase inhibitors, the statement in lines 22-23 of P. 11 needs to be qualified. While the data indicate that inhibition of some kinase blocks syk and CARD9 co-localization, it cannot be definitively concluded that this is due to inhibition of EPHB2.

We are thankful for the reviewer's comment, and we changed the description to "Vandetanib and Dasatinib" in the lines 5-6 of P. 11 in the revised manuscript.

We added one sentence "Due to the lack of specificity of Vandetanib and Dasatinib towards EPHB2, we can't exclude the possibility that the effect of Vandetanib and Dasatinib may partially be due to inhibition of other kinases" in the lines 2-4 of P. 11 in the revised manuscript.

Similarly, the first sentence in the legend of Fig. 7 is not accurate because it cannot be determined whether the effects of the drugs are due to inhibition of EPHB2 or some other kinase.

We changed the description to “Vandetanib and Dasatinib attenuate EAE severity” for the head of Fig. 8 in the revised manuscript, which is Fig. 7 in the first version.

4. Again, the statement on lines 12-13 on P. 13 is not accurate and needs to be qualified to state that the dasatanib and vendetanib do not definitely link inhibition of EPHB2 to the outcome of EAE because they inhibit so many kinases.

We are thankful for the reviewer's comment, and we changed the description to “two existing drugs Vandetanib and Dasatinib” in the lines 23-24 of P. 12 in the revised manuscript.

5. Throughout the manuscript, the authors use the words, “anti-fungi signaling.” The correct terminology is “anti-fungal signaling.”

We are sorry for the mistake, and we corrected it throughout the manuscript.

6. The Western blots in Figs. 1G and S2A are not of publication quality. In Fig. 1G it is difficult for this reviewer to see if there is any significant phosphorylation of Raf or syk in any of the cells at any time point. In Fig. S2A, the phosphorylation of ERK appears to be constitutive.

We are thankful for the reviewer's comment, and we repeated these two experiments, and got better representative data, and please find them in Fig. 1g and Supplementary Figure 2A in the revised manuscript.

Different from the pattern of p-SYK, we found that P-Raf-1 is super-shifted after heat killed *C. albicans* stimulation, while the extent of shift was comparable between control and TAGAP-KO BMDMs.

7. P. 8 line 13, “medicated” should be changed to “mediated.”

We are sorry for the mistake, and we corrected it in the revised manuscript.